# Skill Machines: Temporal Logic Skill Composition in Reinforcement Learning

**Geraud Nangue Tasse, Devon Jarvis, Steven James & Benjamin Rosman**
School of Computer Science and Applied Mathematics
University of the Witwatersrand
Johannesburg, South Africa
{geraud.nanguetasse1, devon.jarvis, steven.james, benjamin.rosman1}@wits.ac.za

## Abstract

It is desirable for an agent to be able to solve a rich variety of problems that can be specified through language in the same environment. A popular approach towards obtaining such agents is to reuse skills learned in prior tasks to generalise compositionally to new ones. However, this is a challenging problem due to the curse of dimensionality induced by the combinatorially large number of ways high-level goals can be combined both logically and temporally in language. To address this problem, we propose a framework where an agent first learns a sufficient set of skill primitives to achieve all high-level goals in its environment. The agent can then flexibly compose them both logically and temporally to provably achieve temporal logic specifications in any regular language, such as regular fragments of linear temporal logic. This provides the agent with the ability to map from complex temporal logic task specifications to near-optimal behaviours zero-shot. We demonstrate this experimentally in a tabular setting, as well as in a high-dimensional video game and continuous control environment. Finally, we also demonstrate that the performance of skill machines can be improved with regular off-policy reinforcement learning algorithms when optimal behaviours are desired.

## 1 Introduction

While reinforcement learning (RL) has achieved recent success in several applications, ranging from video games (Badia et al., 2020) to robotics (Levine et al., 2016), there are several shortcomings that hinder RL's real-world applicability. One issue is that of sample efficiency—while it is possible to collect millions of data points in a simulated environment, it is simply not feasible to do so in the real world. This inefficiency is exacerbated when a single agent is required to solve multiple tasks, as we would expect of a generally intelligent agent.

One approach to overcoming this challenge is to reuse learned behaviours to solve new tasks (Taylor & Stone, 2009), preferably without further learning. Such an approach is often *compositional*— an agent first learns individual skills and then combines them to produce novel behaviours. There are several notions of compositionality in the literature, such as *spatial* composition (Todorov, 2009; Van Niekerk et al., 2019), where skills are combined to produce a new single behaviour to be executed to achieve sets of high-level goals ("pick up an object that is both blue and a box"), and *temporal* composition (Sutton et al., 1999; Jothimurugan et al., 2021), where sub-skills are invoked one after the other to achieve sequences of high-level goals (for example, "pickup a blue object and then a box").

Spatial composition is commonly achieved through a weighted combination of learned successor features (Barreto et al., 2018; 2019; Alver & Precup, 2022). Notably, work by Nangue Tasse et al. (2020; 2022b) has demonstrated spatial composition using Boolean operators, such as negation and conjunction, producing semantically meaningful behaviours without further learning. This ability can then be leveraged by agents to follow natural language instructions (Cohen et al., 2021; 2022).

One of the most common approaches to temporal composition is to learn options for achieving the sub-goals present in temporal logic tasks while learning a high-level policy over the options to actually solve the task, then reusing the learned options in new tasks (Araki et al., 2021; Icarte et al.,

2022). However, other works like Vaezipoor et al. (2021) have proposed end-to-end neural network architectures for learning sub-skills from a training set that can generalise to similar new tasks.

Liu et al. (2022) observe that for all these prior works, some of the sub-skills (e.g., options) learned from previous tasks can not be transferred satisfactorily to new tasks and provide a method to determine when this is the case. For example, if the agent has previously learned an option for "getting blue objects" and another for "getting boxes", it can reuse them to "pickup a blue object and then a box", but it cannot reuse them to "pickup a blue object that is not a box, and then a box that is not blue". We can observe that this problem is because all the compositions in prior works are *either strictly temporal or strictly spatial*. While the example shows that temporal composition alone is insufficient, notice that spatial composition is also not enough for solving long-horizon tasks. In these instances, it is often near impossible for the agent to learn, owing to the large sequence of actions that must be executed before a learning signal is received (Arjona-Medina et al., 2019).

Hence, this work aims to address the highlighted problem by combining the approaches above to develop an agent capable of both zero-shot spatial *and* temporal composition. We particularly focus on temporal logic composition, such as linear temporal logic (LTL) (Pnueli, 1977), allowing agents to sequentially chain and order their skills while ensuring certain conditions are always or never met. We make the following main contributions:

1. **Skill machines:** We propose *skill machines (SM)*, which are finite state machines (FSM) that encode the solution to any task specified using any given regular language (such as regular fragments of LTL) as a series of Boolean compositions of *skill primitives*—composable sub-skills for achieving high-level goals in the environment. An SM is defined by translating the regular language task specification into an FSM, and defining the skill to use per FSM state as a Boolean composition of pretrained skill primitives.

2. **Zero-shot and few-shot learning using skill machines:** By leveraging reward machines (RM) (Icarte et al., 2018a)—finite state machines that encode the reward structure of a task—we show how an SM can be obtained directly from an LTL task specification, and prove that these SMs are *satisficing*—given a task specification and regular reachability assumptions, an agent can successfully solve the task while adhering to any constraints. We further show how standard off-policy RL algorithms can be used to improve the resulting behaviours when optimality is desired. This is achieved with no new assumption in RL.

3. **Emperical and qualitative results:** We demonstrate our approach in several environments, including a high-dimensional video game and a continuous control environment. Our results indicate that our method is capable of producing near-optimal to optimal behaviour for a variety of long-horizon tasks without further learning, including empirical results that far surpass all the representative state-of-the-art baselines.

## 2 BACKGROUND

We model the agent's interaction with the world as a Markov Decision Process (MDP), given by $(\mathcal{S}, \mathcal{A}, \rho, R, \gamma)$, where (i) $\mathcal{S}$ is the finite set of all states the agent can be in; (ii) $\mathcal{A}$ is the finite set of actions the agent can take in each state; (iii) $\rho(s'|s, a)$ is the dynamics of the world; (iv) $R : \mathcal{S} \times \mathcal{A} \times \mathcal{S} \rightarrow \mathbb{R}$ is the reward function; (v) $\gamma \in [0, 1]$ is a discount factor. The agent's aim is to compute a Markov policy $\pi$ from $\mathcal{S}$ to $\mathcal{A}$ that optimally solves a given task. Instead of directly learning a policy, an agent can instead learn a value function that represents the expected return of executing an action $a$ from a state $s$, and then following $\pi$: $Q^\pi(s, a) = \mathbb{E}^\pi \left[\sum_{t=0}^{\infty} \gamma^t R(s_t, a_t, s_{t+1})\right]$. The optimal action-value function is given by $Q^*(s, a) = \max_\pi Q^\pi(s, a)$ for all states $s$ and actions $a$, and the optimal policy follows by acting greedily with respect to $Q^*$ at each state: $\pi^*(s) \in \arg\max_a Q^*(s, a)$.

### 2.1 LTL AND REWARD MACHINES

One difficulty with the standard MDP formulation is that the agent is often required to solve a complex long-horizon task using only a scalar reward signal as feedback from which to learn. To overcome this, a common approach is to use reward machines (RM) (Icarte et al., 2018b), which provide structured feedback to the agent in the form of a finite state machine (FSM). Camacho et al. (2019) show that temporal logic tasks specified using regular languages, such as regular fragments of

LTL (like safe, co-safe, and finite trace LTL), can be converted to RMs with rewards of 1 for accepting transitions and 0 otherwise (Figure 1 shows an example).[1] Hence, without loss of generality, we will focus our attention on tasks specified using regular fragments of LTL—such as co-safe LTL (Kupferman & Vardi, 2001). These LTL specifications and RMs encode the task to be solved using a set of propositional symbols $\mathcal{P}$ that represent high-level environment features as follows:

**Definition 2.1** (LTL). An LTL expression is defined using the following recursive syntax: $\varphi := p \mid \neg\varphi \mid \varphi_1 \vee \varphi_2 \mid \varphi_1 \wedge \varphi_2 \mid X\varphi \mid G\varphi \mid \varphi_1 U \varphi_2 \mid \varphi_1 F \varphi_2$, where $p \in \mathcal{P}$; $\neg$ (*not*), $\vee$ (*or*), $\wedge$ (*and*) are the usual Boolean operators; $X$ (*neXt*), $G$ (*Globally* or *always*), $U$ (*Until*), $F$ (*Finally* or *eventually*) are the LTL temporal operators; and $\varphi, \varphi_1, \varphi_2$ are any valid LTL expression.

**Definition 2.2** (RM). Given a set of environment states $\mathcal{S}$ and actions $\mathcal{A}$, a reward machine is a tuple $R_{\mathcal{S}\mathcal{A}} = \langle \mathcal{U}, u_0, \delta_u, \delta_r \rangle$ where (i) $\mathcal{U}$ is a finite set of states; (ii) $u_0 \in \mathcal{U}$ is the initial state; (iii) $\delta_u : \mathcal{U} \times 2^{\mathcal{P}} \to \mathcal{U}$ is the state-transition function; and (iv) $\delta_r : \mathcal{U} \times 2^{\mathcal{P}} \to \{0, 1\}$ is the state-reward function.[2]

To incorporate RMs into the RL framework, the agent must be able to determine a correspondence between abstract RM propositions and states in the environment. To achieve this, the agent is equipped with a labelling function $L : \mathcal{S} \to 2^{\mathcal{P}}$ that assigns truth values to each state the agent visits in its environment. The agent's aim now is to learn a policy $\pi : \mathcal{S} \times \mathcal{U} \to \mathcal{A}$ that maximises the rewards from an RM while acting in an environment $\langle \mathcal{S}, \mathcal{A}, \rho, \gamma, \mathcal{P}, L \rangle$. However, the rewards from the reward machine are not necessarily Markov with respect to the environment. Icarte et al. (2022) shows that a **product MDP** (Definition 2.3 below) between the environment and a reward machine guarantees that the rewards are Markov such that the policy can be learned with standard algorithms such as $Q$-learning. This is because the product MDP uses the cross-product to consolidate how actions in the environment result in simultaneous transitions in the environment and state machine. Thus, product MDPs take the form of standard, learnable MDPs. In the rest of this work, we will refer to these product MDPs as *tasks*. To ensure that the optimal policy is also the policy that maximises the probability of satisfying the temporal logic task specification, we will henceforth assume that the environment dynamics are deterministic.

**Definition 2.3** (Tasks). Let $\langle \mathcal{S}, \mathcal{A}, \rho, \gamma, \mathcal{P}, L \rangle$ represent the environment and $\langle \mathcal{U}, u_0, \delta_u, \delta_r \rangle$ be an RM representing the task rewards. Then a task is a product MDP $M_{\mathcal{T}} = \langle \mathcal{S}_{\mathcal{T}}, \mathcal{A}, \rho_{\mathcal{T}}, R_{\mathcal{T}}, \gamma \rangle$ between the environment and the RM, where $\mathcal{S}_{\mathcal{T}} := \mathcal{S} \times \mathcal{U}$, $R_{\mathcal{T}}(\langle s, u \rangle, a, \langle s', u' \rangle) := \delta_r(u, l')$, $\rho_{\mathcal{T}}(\langle s, u \rangle, a) := \langle s', u' \rangle$, $s' \sim \rho(\cdot | s, a)$, $u' = \delta_u(u, l')$, and $l' = L(s')$.

## 2.2 LOGICAL SKILL COMPOSITION

Consider the multitask setting where for each task $M$, an agent is required to reach some terminal goal states in a goal space $\mathcal{G} \subseteq \mathcal{S}$. Nangue Tasse et al. (2020; 2022a) develop a framework for this setting that allows agents to apply the Boolean operations $\wedge$, $\vee$ and $\neg$ over the space of tasks and value functions. This is achieved by first defining a goal-oriented reward function $\mathbf{R}_M(s, g, a)$ that extends the task rewards $R_M(s, a)$ to penalise an agent for achieving goals different from the one it wished to achieve: $\mathbf{R}_M(s, g, a) := R_{\text{MIN}}$ **if** ($g \neq s$ and $s$ is terminal) **else** $R_M(s, a)$; where $R_{\text{MIN}}$ is the lower bound of the reward function. Using $\mathbf{R}_M(s, g, a)$, the related goal-oriented value function can be defined as $\mathbf{Q}_M^{\bar{\pi}_M}(s, g, a) = \mathbb{E}^{\bar{\pi}_M}[\sum_{t=0}^{\infty} \gamma^t \mathbf{R}_M(s_t, g, a_t)]$. Despite the modification of the regular RL objective, an agent can always recover the regular optimal policy of the given task by maximising over goals and actions: $\pi_M^*(s) \in \arg\max_a \max_g \mathbf{Q}_M^*(s, g, a)$.

If a new task can be represented as the logical expression of previously learned tasks, and all tasks differ only in their rewards at goal states (that is, all tasks share the same state and action space, transition dynamics, discount factor, and non-terminal rewards), Nangue Tasse et al. (2022a) prove that the optimal policy can immediately be obtained by composing the learned goal-oriented value functions using the same expression. For example, the $\vee$, $\wedge$, and $\neg$ of two goal-reaching tasks $A$ and $B$ can respectively be solved as follows (we omit the value functions' parameters for readability):

$$\mathbf{Q}_A^* \vee \mathbf{Q}_B^* = \max\{\mathbf{Q}_A^*, \mathbf{Q}_B^*\}; \quad \mathbf{Q}_A^* \wedge \mathbf{Q}_B^* = \min\{\mathbf{Q}_A^*, \mathbf{Q}_B^*\}; \quad \neg\mathbf{Q}_A^* = (\mathbf{Q}_{MAX}^* + \mathbf{Q}_{MIN}^*) - \mathbf{Q}_A^*;$$

where $\mathbf{Q}_{MAX}^*$ and $\mathbf{Q}_{MIN}^*$ are the goal-oriented value functions for the maximum task ($R_{\tau} = R_{\text{MAX}}$ for all $\mathcal{G}$) and minimum task ($R_{\tau} = R_{\text{MIN}}$ for all $\mathcal{G}$), respectively. Following Nangue Tasse et al. (2022b), we will also refer to these goal-oriented value functions as *world value functions* (WVFs).

---

[1] Accepting transitions are those at which the high-level task—described, for example, by LTL—is satisfied.
[2] RMs are more general, but for clarity, we focus on the subset that is obtained from regular languages.

# 3 SKILL COMPOSITION FOR TEMPORAL LOGIC TASKS

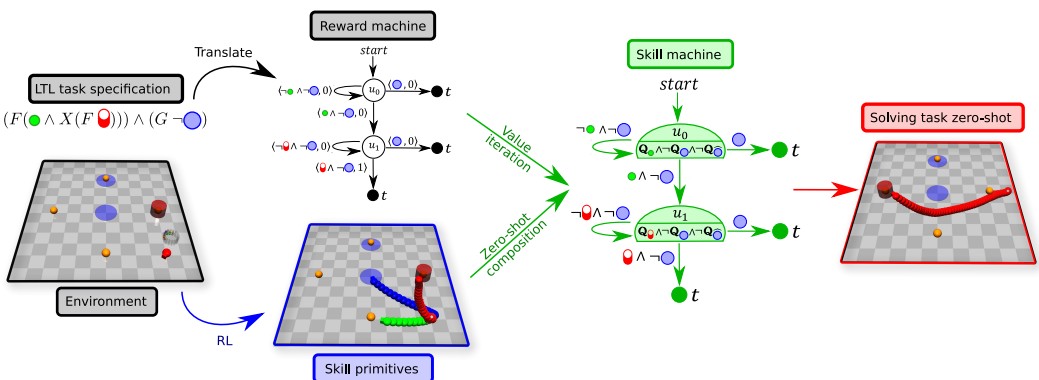

Figure 1: Illustration of our framework: Consider a continuous environment containing a robot (*red sphere*) with 3 LiDAR sensors that it uses to sense when it has reached a red cylinder (), a green button (), or a blue region (). The agent first learns *skill primitives* to reach these 3 objects (the red, green, and blue sample trajectories obtained from them respectively). Then given any task specification over these 3 objects, such as: "Navigate to a button and then to a cylinder while never entering blue regions" with LTL specification $(F(\bullet \land X(F\,\textcolor{red}{\blacksquare}))) \land (G\,\neg\textcolor{blue}{\bigcirc})$, the agent first translates the *LTL task specification* into an RM, then plans which spatial skill to use at each temporal node using *value iteration* and composes its skill primitives to obtain said spatial skills (culminating in a *skill machine*), and finally uses them to solve the task without further learning. The RM is obtained by converting the LTL expression into an FSM using Spot (Duret-Lutz et al., 2016), then giving a reward of $1$ for accepting transitions and $0$ otherwise. The nodes labeled $t$ in the RM and SM represent terminal states (sink/absorbing states where no transition leaves the state).

To describe our approach, we use the *Safety Gym Domain* (Ray et al., 2019) shown in Figure 1 as a running example. Here, the agent moves by choosing a direction and force ($\mathcal{A} = \mathbb{R}^2$) and observes a real vector containing various sensory information like joint velocities and distance to the objects in its surrounding ($\mathcal{S} = \mathbb{R}^{60}$). The LTL tasks in this environment are defined over 3 propositions: $\mathcal{P} = \{\textcolor{red}{\blacksquare}, \bullet, \textcolor{blue}{\bigcirc}\}$, where each proposition is true when the agent is $\epsilon = 1$ metre near its respective location.

Now consider an agent that has learned how to "Go to the cylinder" ($F\,\textcolor{red}{\blacksquare}$), "Go to a button" ($F\,\bullet$), and "Go to a blue region" ($F\,\textcolor{blue}{\bigcirc}$). Say the agent is now required to solve the task with LTL specification $(F(\bullet \land X(F\,\textcolor{red}{\blacksquare}))) \land (G\,\neg\textcolor{blue}{\bigcirc})$. Using prior LTL transfer works (Vaezipoor et al., 2021; Jothimurugan et al., 2021; Liu et al., 2022), the agent would have learned options for solving the first 3 tasks, but then would be unable to transfer those skills to immediately solve this new task. This is because the new task requires the agent to first reach a button that is not in a blue region (*eventually* satisfy $\bullet \land \neg\textcolor{blue}{\bigcirc}$) while not entering a blue region along the way (*always* satisfy $\neg\textcolor{blue}{\bigcirc}$). Similarly, it then must eventually satisfy $\textcolor{red}{\blacksquare} \land \neg\textcolor{blue}{\bigcirc}$ while never satisfying $\textcolor{blue}{\bigcirc}$. However, all 3 options previously learned will enter a blue region if it is along the agent's path. Hence the agent will need to learn new options for achieving $\bullet \land \neg\textcolor{blue}{\bigcirc}$ and $\textcolor{red}{\blacksquare} \land \neg\textcolor{blue}{\bigcirc}$ where the option policies never enter $\textcolor{blue}{\bigcirc}$ along the way.

In general, we can see that there are $2^{2^{\mathcal{P}}}$ possible Boolean expressions the agent may be required to *eventually* satisfy (spatial curse), and $2^{2^{\mathcal{P}}}$ possible Boolean expressions the agent may be required to *always* satisfy (temporal curse). This highlights the curses of dimensionality we aim to simultaneously address. In this section, we will introduce skill primitives as the proposed solution for addressing the aforementioned curses of dimensionality. We will then introduce skill machines as a state machine that can encode the solution to any temporal logic task by leveraging skill primitives.

## 3.1 FROM ENVIRONMENT TO PRIMITIVES

We desire an agent capable of learning a sufficient set of skills that can be used to solve new tasks, specified through LTL, with little or no additional learning. To achieve this, we introduce the notion of *primitives* which aims to address the spatial and temporal curses of dimensionality as follows:

**Spatial curse of dimensionality:**   To address this, we can learn WVFs (the composable value functions described in Section 2.2) for *eventually* achieving each proposition, then compose them to *eventually* achieve the Boolean expression over the propositions. For example, we can learn WVFs for tasks $F$ 🔴, $F$ 🟢, and $F$ 🔵. However, the product MDP for LTL specified tasks have different states and dynamics (see Definition 2.3). Hence, they do not satisfy the assumptions for zero-shot logical composition (Section 2.2). To address this problem, we define task primitives below. These are product MDPs for achieving each proposition when the agent decides to terminate, and share the same state space and dynamics. We then define skill primitives as their corresponding WVFs.

**Temporal curse of dimensionality:**   To address this, we introduce the concept of *constraints* $\mathcal{C} \subseteq \{\widehat{p} \mid p \in \mathcal{P}\}$ which we use to augment the state space of task primitives[3]. In a given environment, a constraint is a proposition that an agent may be required to *always* keep True or *always* keep False in some FSM state of a temporal logic task. Equivalently, it is a proposition which may never change across the trajectory of the agent in the FSM state. When contradicted it may transition the agent into a failure FSM state (an FSM sink state from which it can never solve the task). For example, some tasks like $(F(🟢 \wedge X(F\ 🔴))) \wedge (G\ \neg 🔵)$ require the agent to solve a task $F(🟢 \wedge X(F\ 🔴))$ while never setting 🔵 to True ($G\ \neg 🔵$). By setting the 🔵 proposition as a constraint when learning a primitive (e.g achieving 🟢), the agent keeps track (in its cross-product state) of whether or not it has reached a blue region in a trajectory that did not start in a blue region. That is, in an episode where the agent does not start in a blue region but later goes through a blue region and terminates at a button, the agent will achieve the goal $g = \{🟢, \widehat{🔵}\} \in 2^{\mathcal{P} \cup \mathcal{C}}$. We henceforth assume the general case $\mathcal{C} = \{\widehat{p} \mid p \in \mathcal{P}\}$ for our theory, then later consider different choices for $\mathcal{C}$ in our experiments.

We now formally define the notions of task primitives and skill primitives such as "Go to a button":

**Definition 3.1** (Primitives). Let $\langle \mathcal{S}, \mathcal{A}, \rho, \gamma, \mathcal{P}, L \rangle$ represent the environment the agent is in, and $\mathcal{C}$ be the set of constraints. We define a *task primitive* here as an MDP $M_p = \langle \mathcal{S}_\mathcal{G}, \mathcal{A}_\mathcal{G}, \rho_\mathcal{G}, R_p, \gamma \rangle$ with absorbing states $\mathcal{G} = 2^{\mathcal{P} \cup \mathcal{C}}$ that corresponds to achieving a proposition $p \in \mathcal{P} \cup \mathcal{C}$, where $\mathcal{S}_\mathcal{G} := (\mathcal{S} \times 2^\mathcal{C}) \cup \mathcal{G}$; $\mathcal{A}_\mathcal{G} := \mathcal{A} \times \mathcal{A}_\tau$, where $\mathcal{A}_\tau = \{0, 1\}$ is an action that terminates the task;

$$\rho_\mathcal{G}(\langle s, c\rangle, \langle a, a_\tau \rangle) := \begin{cases} l' \cup c & \text{if } a_\tau = 1 \\ \langle s', c' \rangle & \text{otherwise} \end{cases} ;\ R_p(\langle s, c\rangle, \langle a, a_\tau \rangle) := \begin{cases} 1 & \text{if } a_\tau = 1 \text{ and } p \in l' \cup c \\ 0 & \text{otherwise} \end{cases},$$

where $s' \sim \rho(\cdot|s, a)$, $l = L(s)$, $l' = L(s')$, and $c' = c \cup ((\widehat{l} \oplus \widehat{l'}) \cap \mathcal{C})$.

A *skill primitive* is defined as $\mathbf{Q}_p^*(\langle s, c\rangle, g, \langle a, a_\tau \rangle)$, the WVF for the task primitive $M_p$.

The above defines the state space of primitives to be the product of the environment states and the set of constraints, incorporating the set of propositions that are currently true. The action space is augmented with a terminating action following Barreto et al. (2019) and Nangue Tasse et al. (2020), which indicates that the agent wishes to achieve the goal it is currently at, and is similar to an option's termination condition (Sutton et al., 1999). The transition dynamics update the environment state $s$ and the set of violated constraints $c$ when any other action is taken. Here, the labeling function is used to return the set of propositions $l$ and $l'$ achieved in $s$ and $s'$ respectively. Any constraint present exclusively in $l$ or $l'$ is added to $c$, since it has not been kept always True or always False. Finally, the agent receives a reward of $1$ when it terminates in a state where the proposition $p$ is true, and $0$ otherwise. Figure A7 shows examples of the resulting optimal policies when the set of constraints is empty and non-empty.

Since all task primitives $\mathcal{M}_\mathcal{G} := \{M_p \mid p \in \mathcal{P} \cup \mathcal{C}\}$ share the same state space, action space, dynamics, and rewards at non-terminal states, the corresponding skill primitives $\mathbfcal{Q}_\mathcal{G}^* := \{\mathbf{Q}_p^* \mid p \in \mathcal{P} \cup \mathcal{C}\}$ can be composed to achieve any Boolean expression over $\mathcal{P} \cup \mathcal{C}$ (Nangue Tasse et al., 2022a). We next introduce *skill machines* which leverages skill primitives to encode the solution to temporal logic tasks.

## 3.2   SKILL MACHINES

We now have agents capable of solving any logical composition of task primitives $\mathcal{M}_\mathcal{G}$ by learning only their corresponding skill primitives $\mathbfcal{Q}_\mathcal{G}^*$ and using the zero-shot composition operators (Section 2.2). Given this compositional ability over skills, and reward machines that expose the reward structure

---

[3]The notation $\widehat{p}$ represents when a *literal* (a proposition $p \in \mathcal{P}$ or its negation $\neg p$) is being used as a constraint. Similarly, we will use $\widehat{\mathcal{P}}$ or $\widehat{\sigma}$ respectively when the literals in a set $\mathcal{P}$ or Boolean expression $\sigma$ are constraints.

of tasks, agents can solve temporally extended tasks with little or no further learning. To achieve this, we define a skill machine (SM) as a representation of logical and temporal knowledge over skills.

**Definition 3.2** (Skill Machine). Let $\langle \mathcal{S}, \mathcal{A}, \rho, \gamma, \mathcal{P}, L \rangle$ represent the environment the agent is in, and $\mathcal{Q}_{\mathcal{G}}^*$ be the corresponding skill primitives with constraints $\mathcal{C}$. Given a reward machine $R_{\mathcal{SA}} = \langle \mathcal{U}, u_0, \delta_u, \delta_r \rangle$, a skill machine is a tuple $\mathcal{Q}_{\mathcal{SA}}^* = \langle \mathcal{U}, u_0, \delta_u, \delta_Q \rangle$ where $\delta_Q : U \to [\mathcal{S}_{\mathcal{G}} \times \mathcal{A}_{\mathcal{G}} \to \mathbb{R}]$ is the state-skill function defined by:

$$\delta_Q(u)(\langle s, c \rangle, \langle a, 0 \rangle) := \max_{g \in \mathcal{G}} \mathbf{Q}_{\sigma_u}^*(\langle s, c \rangle, g, \langle a, 0 \rangle),$$

and $\mathbf{Q}_{\sigma_u}^*$ is the composition of skill primitives $\mathcal{Q}_{\mathcal{G}}^*$ according to a Boolean expression $\sigma_u \in 2^{2^{\mathcal{P} \cup \mathcal{C}}}$.

For a given state $s \in \mathcal{S}$ in the environment, the set of constraints violated $c \subseteq \mathcal{C}$, and state $u$ in the skill machine, the skill machine computes a skill $\delta_Q(u)(\langle s, c \rangle, \langle a, 0 \rangle)$ that an agent can use to take an action $a$. The environment then transitions to the next state $s'$ with true propositions $l'$—where $\langle s', c' \rangle \leftarrow P_{\mathcal{G}}(\langle s, c \rangle, \langle a, 0 \rangle)$ and $l' \leftarrow L(s')$—and the skill machine transitions to $u' \leftarrow \delta_u(u, l')$. This process is illustrated in Figure A8 for the skill machine shown in Figure 1. Remarkably, because the Boolean compositions of skill primitives are optimal, there always exists a choice of skill machine that is optimal with respect to the corresponding reward machine, as shown in Theorem 3.3, which demonstrates that SMs can be used to solve tasks without having to relearn action level policies:

**Theorem 3.3.** *Let $\pi^*(s, u)$ be the optimal policy for a task $M_{\mathcal{T}}$ specified by an RM $R_{\mathcal{SA}}$. Then there exists a corresponding skill machine $\mathcal{Q}_{\mathcal{SA}}^*$ such that $\pi^*(s, u) \in \arg\max_{a \in \mathcal{A}} \delta_Q(u)(\langle s, c \rangle, \langle a, 0 \rangle)$.*

### 3.3 FROM REWARD MACHINES TO SKILL MACHINES

In the previous section, we introduced skill machines and showed that they can be used to represent the logical and temporal composition of skills needed to solve tasks specified by reward machines. However, we only proved their existence—for a given task, how can we acquire an SM that solves it?

**Zero-shot via planning over the RM:** To obtain the SM that solves a given RM, we first plan over the reward machine (using value iteration, for example) to produce action-values for each transition. We then select skills for each SM state greedily by applying Boolean composition to skill primitives according to the Boolean expressions defining: (i) the transition with the highest value (propositions to eventually satisfy); and (ii) the transitions with zero value (constrains to always satisfy). This process is illustrated by Figure A9. Since the skills per SM state are selected greedily, the policy generated by this SM is *recursively optimal* (Hutsebaut-Buysse et al., 2022)—that is, it is locally optimal (optimal for each sub-task) but may not be globally optimal (optimal for the overall task). Interestingly, we show in Theorem 3.4 that this policy is also *satisficing* (reaches an accepting state) if we assume global reachability—all FSM transitions (that is all Boolean expressions $\sigma \in 2^{2^{\mathcal{P}}}$) are achievable from any environment state. This is a more relaxed version of the assumption "any state is reachable from any other state" that is required to prove optimality in most RL algorithms, since an agent cannot learn an optimal policy if there are states it can never reach.

**Theorem 3.4.** *Let $R_{\mathcal{SA}} = \langle \mathcal{U}, u_0, \delta_u, \delta_r \rangle$ be a satisfiable RM where all the Boolean expressions $\sigma$ defining its transitions are in negation normal form (NNF) (Robinson & Voronkov, 2001) and are achievable from any state $s \in \mathcal{S}$. Define the corresponding SM $\mathcal{Q}_{\mathcal{SA}}^* = \langle \mathcal{U}, u_0, \delta_u, \delta_Q \rangle$ with $\delta_Q(u)(\langle s, c \rangle, \langle a, 0 \rangle) \mapsto \max_{g \in \mathcal{G}} \mathbf{Q}_{(\sigma_{\mathcal{P}} \wedge \neg \sigma_{\mathcal{C}})}^*(\langle s, c \rangle, g, \langle a, 0 \rangle)$ where $\sigma_{\mathcal{P}} := argmax_{\sigma} Q^*(u, \sigma)$, $\sigma_{\mathcal{C}} := \bigvee \{\widehat{\sigma} \mid Q^*(u, \sigma) = 0\}$, and $Q^*(u, \sigma)$ is the optimal Q-function for $R_{\mathcal{SA}}$. Then, $\pi(s, u) \in \arg\max_{a \in \mathcal{A}} \delta_Q(u)(\langle s, c \rangle, \langle a, 0 \rangle)$ is satisficing.*

Theorem 3.4 is critical as it provides soundness guarantees, ensuring that the policy derived from the skill machine will always satisfy the task requirements.

**Few-shot via RL in the environment:** Finally, in cases where the composed skill $\delta_Q(u)(\langle s, c \rangle, \langle a, 0 \rangle)$ obtained from the approximate SM is not sufficiently optimal, we can use any off-policy RL algorithm to learn the task-specific skill $Q_{\mathcal{T}}(s, u, a)$ few-shot. This is achieved by using the maximising Q-values $\max\{\gamma Q_{\mathcal{T}}, (1 - \gamma)\delta_Q\}$ in the exploration policy during learning. Here, the discount factor $\gamma$ determines how much of the composed policy to use. Consider Q-learning, for example: during the $\epsilon$-greedy exploration, we use $a \leftarrow \arg\max_{\mathcal{A}} \max\{\gamma Q_{\mathcal{T}}, (1 - \gamma)\delta_Q\}$ to select greedy actions. This improves the initial performance of the agent where $\gamma Q_{\mathcal{T}} < (1 - \gamma)\delta_Q$, and guarantees convergence in the limit of infinite exploration, as in vanilla Q-learning. Appendix A.2 illustrates this process.

## 4 EXPERIMENTS

We evaluate our approach in three domains, including a high-dimensional, continuous control task. In particular, we consider the Office Gridworld (Figure A2a), the Moving Targets domain (Figure A1) and the Safety Gym domain (Figure 1). We briefly describe the domains and training procedure here, and provide more detail and hyperparameter settings in the appendix.

**Office Gridworld (Icarte et al., 2022):** Tasks are specified over 10 propositions $\mathcal{P} = \{A, B, C, D, ✽, ☕, ✉, 🚶, ✉^+, 🚶^+\}$ and 1 constraint $\mathcal{C} = \{✽\}$. We learn the skill primitives $\mathcal{Q}_{\mathcal{G}}^*$ (visualised by Figure A3) using goal-oriented Q-learning (Nangue Tasse et al., 2020), where the agent keeps track of reached goals and uses Q-learning (Watkins, 1989) to update the WVF with respect to all previously seen goals at every time step.

**Moving Targets Domain (Nangue Tasse et al., 2020):** This is a canonical object collection domain with high dimensional pixel observations ($84 \times 84 \times 3$ RGB images). The agent here needs to pick up objects of various shapes and colours; collected objects respawn at random empty positions similarly to previous object collection domains (Barreto et al., 2020). There are 3 object colours—*beige* (□), *blue* (■), *purple* (■)—and 2 object shapes—*squares* (⊠), *circles* (○). The tasks here are defined over 5 propositions $\mathcal{P} = \{□, ■, ■, ⊠, ○\}$ and 5 constraints $\mathcal{C} = \widehat{\mathcal{P}}$. We learn the corresponding skill primitives with goal-oriented Q-learning, but using deep Q-learning (Mnih et al., 2015) to update the WVFs.

**Safety Gym Domain (Ray et al., 2019):** A continuous state and action space ($\mathcal{S} = \mathbb{R}^{60}, \mathcal{A} = \mathbb{R}^2$) domain where an agent, represented by a point mass, must navigate to various regions defined by 3 propositions ($\mathcal{P} = \{🔴, ●, ⬭\}$) corresponding to its 3 LiDAR sensors for the *red cylinder* 🔴, the *green buttons* ●, and the *blue regions* ⬭. We learn the four skill primitives corresponding to each predicate (with constraints $\mathcal{C} = \{\widehat{⬭}\}$), using goal-oriented Q-learning and TD3 (Fujimoto et al., 2018).

### 4.1 ZERO-SHOT AND FEW-SHOT TEMPORAL LOGICS

| Task | Description — LTL |
|---|---|
| 1 | Deliver coffee to the office without breaking decorations $\mid \left(F\left(☕ \wedge X\left(F\,🚶\right)\right)\right) \wedge (G\,\neg✽)$ |
| 2 | Patrol rooms $A$, $B$, $C$, and $D$ without breaking any decoration $— \left(F\left(A \wedge X\left(F\left(B \wedge X\left(F\left(C \wedge X\left(FD\right)\right)\right)\right)\right)\right)\right) \wedge (G\,\neg✽)$ |
| 3 | Deliver coffee and mail to the office without breaking any decoration $— \left(\left(F\left(☕ \wedge X\left(F\left(⊠ \wedge X\left(F🚶\right)\right)\right)\right)\right) \vee \left(F\left(⊠ \wedge X\left(F\left(☕ \wedge X\left(F🚶\right)\right)\right)\right)\right)\right) \wedge (G\neg✽)$ |
| 4 | Deliver mail to the office until there is no mail left, then deliver coffee to office while there are people in the office, then patrol rooms A-B-C-D-A, and never break a decoration $— \big(F\big(⊠ \wedge X\big(F\big(🚶 \wedge X\big(\neg⊠U\big(\neg⊠^+ \wedge ⊠ \wedge X\big(F\big(☕ \wedge X\big(\neg🚶U\big(\neg🚶^+ \wedge 🚶 \wedge X$ $\left(FA \wedge X\left(F\left(B \wedge X\left(F\left(C \wedge X\left(F\left(D \wedge X\left(FA\right)\right)\right)\right)\right)\right)\right)\right)\big)\big)\big)\big)\big)\big)\big)\big)\big)\big) \wedge (G\,\neg✽)$ |

Table 1: Tasks in the Office Gridworld. The RMs are generated from the LTL expressions.

We use the Office Gridworld as a multitask domain, and evaluate how long it takes an agent to learn a policy that can solve the four tasks described in Table 1. The tasks are sampled uniformly at random for each episode. In all of our experiments, we compare the performance of SMs without further learning and SMs paired with Q-learning (QL-SM) with that of regular Q-learning (QL) and the following state-of-the-art RM-based baselines (Icarte et al., 2022): (i) **Counterfactual RMs (CRM)**: This augments Q-learning by updating the action-value function at each state ($Q(s, u, a)$) not just with respect to the current RM transition, but also with respect to all possible RM transitions from the current environment state. This is representative of approaches that leverage the compositional structure of RMs to learn optimal policies efficiently. (ii) **Hierarchical RMs (HRM)**: The agent here uses Q-learning to learn options to achieve each RM state-transition, and an option policy to select which options to use at each RM state that are grounded in the environment states. This is representative of option-based approaches that learn hierarchically-optimal policies. (iii) **Reward-shaped variants (QL-RS, CRM-RS, HRM-RS)**: The agent here uses the values obtained from value iteration over the RMs for reward shaping, on top of the regular QL, CRM, HRM algorithms. This is representative of approaches that leverage planning over the RM to speed up learning.

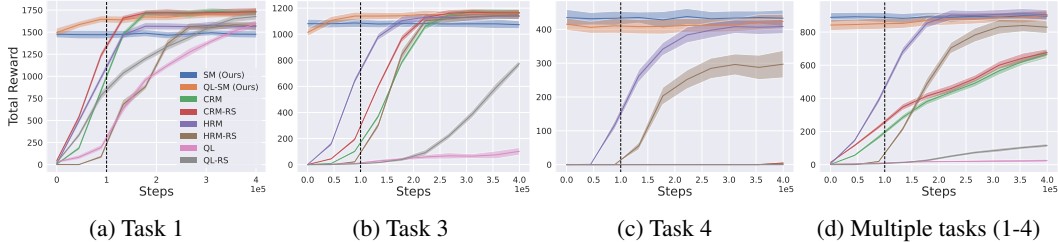

(a) Task 1      (b) Task 3      (c) Task 4      (d) Multiple tasks (1-4)

Figure 2: Average returns over 60 independent runs during training in the Office Gridworld. The shaded regions represent 1 standard deviation. For each training run, we evaluate the agent $\epsilon$-greedily ($\epsilon = 0.1$) after every 1000 step and report the average total rewards obtained over each 40 consecutive evaluation. The black dotted line indicate the point at which the baselines have trained for the same number of time steps as the skill primitives pretraining.

In addition to learning all four tasks at once, we also experiment with Tasks 1, 3 and 4 in isolation. In these single-task domains, the difference between the baselines and our approach should be more pronounced, since QL, CRM and HRM now cannot leverage the shared experience across multiple tasks. Thus, the comparison between multi-task and single-task learning in this setting will evaluate the benefit of the compositionality afforded by SMs, given that the 11 skill primitives used by the SMs here are pretrained only once for $1 \times 10^5$ time steps and used for all four experiments. For fairness towards the baselines, we run each of the four experiments for $4 \times 10^5$ time steps.

The results of these four experiments are shown in Figure 2. Regular Q-learning struggles to learn Task 3 and completely fails to learn the hardest task (Task 4). Additionally, notice that while QL and CRM can theoretically learn the tasks optimally given infinite time, only HRM, SM, and QL-SM are able to learn hard long horizon tasks in practice (like task 4). This is because of the temporal composition of skills leveraged in HRM, SM, and QL-SM. In addition, the skill machines are being used to zero-shot generalise to the office tasks using skill primitives. Thus using the skill machines alone (SM in Figure 2) may provide sub-optimal performance compared to the task-specific agents, since the SMs have not been trained to optimality and are not specialised to the domain. Even under these conditions, we observe that SMs perform near-optimally in terms of final performance, and due to the amortised nature of learning the WVF will achieve its final rewards from the first epoch.

Finally, it is apparent from the results shown in Figure 2 that SMs paired with Q-learning (QL-SM) achieve the best performance when the zero-shot performance is not already optimal (see Appendix A4 for the trajectories of the agent with and without few-shot learning). Additionally, SMs with Q-learning always begin with a significantly higher reward and converge on their final performance faster than all baselines. The speed of learning is due to the compositionality of the skill primitives with SMs, and the high final performance is due to the generality of the learned primitives being paired with the domain-specific Q-learner. In sum, skill machines provide fast composition of skills and achieve optimal performance compared to all benchmarks when paired with a learning algorithm.

## 4.2 ZERO-SHOT TRANSFER WITH FUNCTION APPROXIMATION

We now demonstrate our temporal logic composition approach in the Moving Targets domain where function approximation is required. Figure 3 shows the average returns of the optimal policies and SM policies for the four tasks described in Table 2 with a maximum of 50 steps per episode. Our results show that even when using function approximation with sub-optimal skill primitives, the zero-shot policies obtained from skill machines are very close to optimal on average. We also observe that for very challenging tasks like Tasks 3 and 4 (where the agent must satisfy difficult temporal constraints), the compounding effect of the sub-optimal policies sometimes leads to failures. Finally, we provide a qualitative demonstration of our method's applicability to continuous control tasks using Safety Gym, a benchmark domain used for developing safe RL methods (Ray et al., 2019). We define a set of increasingly complex tasks and visualise the resulting trajectories after composing the agent's learned primitive skills. Figure 1 illustrates the trajectory that satisfies the task requiring the agent to navigate to a blue region, then to a red cylinder, and finally to another red cylinder while avoiding blue regions. See Appendix A.5 for all task specifications and trajectory visualisations.

| Task | Description — LTL |
|------|-------------------|
| 1 | Pick up any object. Repeat this forever. — $F(\bigcirc \vee \boxtimes)$ |
| 2 | Pick up blue then purple objects, then objects that are neither blue nor purple. Repeat this forever. — $F(\blacksquare \wedge X(F(\blacksquare \wedge X(F((\bigcirc \vee \boxtimes) \wedge \neg(\blacksquare \vee \blacksquare))))))$ |
| 3 | Pick up blue objects or squares, but never blue squares. Repeat this forever. — $(F(\blacksquare \vee \boxtimes)) \wedge (G\, \neg(\blacksquare \wedge \boxtimes))$ |
| 4 | Pick up non-square blue objects, then non-blue squares in that order. Repeat this forever. — $F((\neg\boxtimes \wedge \blacksquare) \wedge X(F(\boxtimes \wedge \neg\blacksquare)))$ |

Table 2: Tasks in the Moving Targets domain. To repeat forever, the terminal states of the RMs generated from LTL are removed, and transitions to them are looped back to the start state.

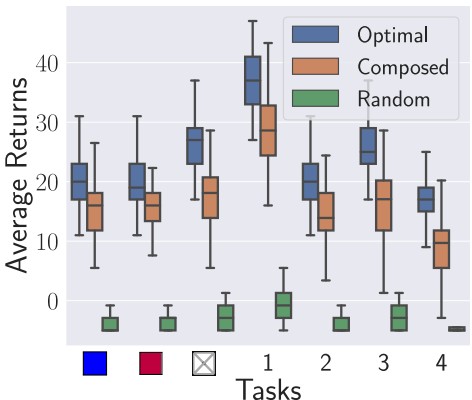

Figure 3: Average returns over 100 runs for tasks in Table 2. The agent and object positions are randomised and objects respawn in random positions when collected.

## 5 RELATED WORK

Regularisation has previously been used to achieve semantically meaningful disjunction (Todorov, 2009; Van Niekerk et al., 2019) or conjunction (Haarnoja et al., 2018; Hunt et al., 2019). Weighted composition has also been demonstrated; for example, Peng et al. (2019) learn weights to compose existing policies multiplicatively to solve new tasks. Approaches built on successor features (SF) are capable of solving tasks defined by linear preferences over features (Barreto et al., 2020)., while Alver & Precup (2022) show that an SF basis can be learned that is sufficient to span the space of tasks under consideration. By contrast, our framework allows for both spatial composition (including operators such as negation that others do not support) and temporal composition such as LTL.

A popular way of achieving temporal composition is through the options framework (Sutton et al., 1999). Here, high-level skills are first discovered and then executed sequentially to solve a task (Konidaris & Barto, 2009). Barreto et al. (2019) leverage the SF and options framework and learn how to linearly combine skills, chaining them sequentially to solve temporal tasks. However, these approaches offer a relatively simple form of temporal composition. By contrast, we are able to solve tasks expressed through regular languages zero-shot, while providing soundness guarantees.

Approaches to defining tasks using human-readable logic operators also exist. Li et al. (2017) and Littman et al. (2017) specify tasks using LTL, which is then used to generate a reward signal for an RL agent. Camacho et al. (2019) perform reward shaping given LTL specifications, while Jothimurugan et al. (2019) develop a formal language that encodes tasks as sequences, conjunctions and disjunctions of subtasks. This is then used to obtain a shaped reward function that can be used for learning. These approaches focus on how to improve learning given such specifications, but we show how an explicitly compositional agent can immediately solve such tasks using WVFs without further learning.

## 6 CONCLUSION

We proposed skill machines—finite state machines that can be learned from reward machines—that allow agents to solve extremely complex tasks involving temporal and spatial composition. We demonstrated how skills can be learned and encoded in a specific form of goal-oriented value function that, when combined with the learned skill machines, are sufficient for solving subsequent tasks without further learning. Our approach guarantees that the resulting policy adheres to the logical task specification, which provides assurances of safety and verifiability to the agent's decision making, important characteristics that are necessary if we are to ever deploy RL agents in the real world. While the resulting behaviour is provably satisficing, empirical results demonstrate that the agent's performance is near optimal; further fine-tuning can be performed should optimality be required, which greatly improves the sample efficiency. We see this approach as a step towards truly generally intelligent agents, capable of immediately solving human-specifiable tasks in the real world with no further learning.

## ACKNOWLEDGEMENTS

Computations were performed using the High Performance Computing Infrastructure provided by the Mathematical Sciences Support unit at the University of the Witwatersrand. G.N.T. is supported by an IBM PhD Fellowship. D.J. is a Google PhD Fellow and Commonwealth Scholar. B.R. is a CIFAR Azrieli Global Scholar in the Learning in Machines & Brains program.

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

## A APPENDIX

### A.1 PROOFS OF THEORETICAL RESULTS

**Theorem A.1.** *Let $\pi^*(s, u)$ be the optimal policy for a task $M_\mathcal{T}$ specified by an RM $R_{\mathcal{SA}}$. Then there exists a corresponding skill machine $\mathcal{Q}^*_{\mathcal{SA}}$ such that*

$$\pi^*(s, u) \in \underset{a \in \mathcal{A}}{\arg\max}\, \delta_Q(u)(\langle s, c \rangle, \langle a, 0 \rangle).$$

*Proof.* Define the skill per SM state $\mathbf{Q}^*_u$ to be the Boolean composition of skill primitives that satisfy the set of propositions $g \in 2^{\mathcal{P} \cup \mathcal{C}}$, where $g$ is the set of propositions satisfied and constraints violated when following $\pi^*(s, u)$. Then $\pi^*(s, u) \in \arg\max_{a \in \mathcal{A}} \delta_Q(u)(\langle s, c \rangle, \langle a, 0 \rangle)$ since $\mathbf{Q}^*_u$ is optimal using Nangue Tasse et al. (2022b) and optimal policies maximise the probability reaching goals (since the rewards are non-zero only at the desirable goal states, where they are 1). $\qquad\square$

**Theorem A.2.** *Let $R_{\mathcal{SA}} = \langle \mathcal{U}, u_0, \delta_u, \delta_r \rangle$ be a satisfiable RM where all the Boolean expressions $\sigma$ defining its transitions are in negation normal form (NNF) (Robinson & Voronkov, 2001) and are achievable from any state $s \in \mathcal{S}$. Define the corresponding SM $\mathcal{Q}^*_{\mathcal{SA}} = \langle \mathcal{U}, u_0, \delta_u, \delta_Q \rangle$ with $\delta_Q(u)(\langle s, c \rangle, \langle a, 0 \rangle) \mapsto \max_{g \in \mathcal{G}} \mathbf{Q}^*_{(\sigma_\mathcal{P} \wedge \neg \sigma_\mathcal{C})}(\langle s, c \rangle, g, \langle a, 0 \rangle)$ where $\sigma_\mathcal{P} \leftarrow argmax_\sigma Q^*(u, \sigma)$, $\sigma_\mathcal{C} \leftarrow \bigvee \{\widehat{\sigma} \mid Q^*(u, \sigma) = 0\}$, and $Q^*(u, \sigma)$ is the optimal Q-function for $R_{\mathcal{SA}}$. Then, $\pi(s, u) \in \arg\max_{a \in \mathcal{A}} \delta_Q(u)(\langle s, c \rangle, \langle a, 0 \rangle)$ is satisficing.*

*Proof.* This follows from the optimality of Boolean skill composition and the optimality of value iteration, since each transition of the RM is satisfiable from any environment state.

$\qquad\square$

### A.2 FULL PSEUDO-CODES OF FRAMEWORK

---

**Algorithm 1:** Q-learning for skill primitives

---

**Input** : $\mathcal{S}, \mathcal{A}, \mathcal{P}, \mathcal{C}, \gamma, \alpha, R_{\text{MAX}} = 1, R_{\text{MIN}} = 0$

**Initialise :** $\mathbf{Q}_{MAX}(\langle s, c \rangle, g, \langle a, a_\tau \rangle)$ and $\mathbf{Q}_{MIN}(\langle s, c \rangle, g, \langle a, a_\tau \rangle)$, goal buffer $G = \{\emptyset\}$

1 **foreach** *episode* **do**
2     Observe initial state $s \in \mathcal{S}$ and true propositions $l \in 2^{\mathcal{P}}$, sample $c \in 2^{\mathcal{C}}$ and $g \in G$
3     **while** *episode is not done* **do**
4        $\langle a, a_\tau \rangle \leftarrow \begin{cases} \underset{\langle a, a_\tau \rangle}{\arg\max} \ \mathbf{Q}_{MAX}(\langle s, c \rangle, g, \langle a, a_\tau \rangle) & \text{if } Bernoulli(1 - \epsilon) = 1 \\ \text{sample } \langle a, a_\tau \rangle \in \mathcal{A} \times \{0, 1\} & \text{otherwise} \end{cases}$
5        Execute $a$ and observe next state $s'$ and true propositions $l'$
6        Get true constraints $c' \leftarrow c \cup ((\widehat{l} \oplus \widehat{l'}) \cap \mathcal{C})$
7        **if** $(a_\tau = 1)$ **then** $G \leftarrow G \ \cup \ \{l' \cup c\}$
8        **foreach** $\mathbf{Q} \in \{\mathbf{Q}_{MAX}, \mathbf{Q}_{MIN}\}$ **do**
9           **if** $(a_\tau \neq 1)$ **then** $r \leftarrow 0$
10          **if** $(a_\tau = 1$ and $\mathbf{Q} = \mathbf{Q}_{MAX})$ **then** $r \leftarrow R_{\text{MAX}}$
11          **if** $(a_\tau = 1$ and $\mathbf{Q} = \mathbf{Q}_{MIN})$ **then** $r \leftarrow R_{\text{MIN}}$
12          **foreach** $g' \in G$ **do**
13             $\bar{r} \leftarrow R_{\text{MIN}}$ **if** $(a_\tau = 1$ and $g' \neq l' \cup c)$ **else** $r$
14             **if** *($s'$ is terminal or $a_\tau = 1$)* **then**
15                $\mathbf{Q}(\langle s, c \rangle, g', \langle a, a_\tau \rangle) \overset{\alpha}{\leftarrow} \bar{r}$
16             **else**
17                $\mathbf{Q}(\langle s, c \rangle, g', \langle a, a_\tau \rangle) \overset{\alpha}{\leftarrow} \left[ \bar{r} + \gamma \max_{\langle a', a'_\tau \rangle} \mathbf{Q}(\langle s', c' \rangle, g', \langle a', a'_\tau \rangle) \right]$
18        $s \leftarrow s'$ and $c \leftarrow c'$
19        **if** $(a_\tau = 1)$ **then** terminate episode
20 $\mathcal{Q}_{\mathcal{G}} \leftarrow \emptyset$
21 **foreach** $p \in P \cup \mathcal{C}$ **do**
22     $\mathbf{Q}_p(\langle s, c \rangle, g, \langle a, a_\tau \rangle) := \mathbf{Q}_{MAX}(\langle s, c \rangle, g, \langle a, a_\tau \rangle)$ **if** $(p \in g)$ **else** $\mathbf{Q}_{MIN}(\langle s, c \rangle, g, \langle a, a_\tau \rangle)$
23     $\mathcal{Q}_{\mathcal{G}} \leftarrow \mathcal{Q}_{\mathcal{G}} \cup \{\mathbf{Q}_p\}$
24 **return** $\mathcal{Q}_{\mathcal{G}}$

---

**Algorithm 2:** Skill machine from reward machine

---

**Input** : $\mathcal{Q}_{\mathcal{G}}, \langle \mathcal{U}, u_0, \delta_u, \delta_r, \rangle, \gamma$

**Initialise :** RM value function $Q(u, \sigma)$, value iteration error $\Delta = 1$

1 Let $\mathcal{B}(u) := $ the set of Boolean expressions defining the RM transitions $\delta_u(u, \cdot)$
    /* Value iteration                                            */
2 **while** $\Delta > 0$ **do**
3     $\Delta \leftarrow 0$
4     **for** $u \in \mathcal{U}$ **do**
5        **for** $\sigma \in \mathcal{B}(u)$ **do**
6           $v' \leftarrow \delta_r(u, \sigma) + \gamma \max_{\sigma'} \ Q(\delta_u(u, \sigma), \sigma')$
7           $\Delta = \max\{\Delta, |Q(u, \sigma) - v'|\}$
8           $Q(u, \sigma) \leftarrow v'$
9
    /* Skill machine's skill function                            */
10 **for** $u \in \mathcal{U}$ **do**
11     $\sigma_{\mathcal{P}}, \sigma_{\mathcal{C}} \leftarrow argmax_{\sigma'} \ Q(u, \sigma'), \ \bigvee\{\widehat{\sigma} \mid Q(u, \sigma) = 0\}$
12     $\mathbf{Q}_{\sigma_{\mathcal{P}} \wedge \neg \sigma_{\mathcal{C}}} \leftarrow$ composition of $\mathcal{Q}_{\mathcal{G}}$ as per the Boolean expression $\sigma_{\mathcal{P}} \wedge \neg \sigma_{\mathcal{C}}$
13     $\delta_Q(u)(\langle s, c \rangle, \langle a, 0 \rangle) \leftarrow \max_{g \in \mathcal{G}} \mathbf{Q}_{\sigma_{\mathcal{P}} \wedge \neg \sigma_{\mathcal{C}}}(\langle s, c \rangle, g, \langle a, 0 \rangle)$
14 **return** $\langle \mathcal{U}, u_0, \delta_u, \delta_Q \rangle$

---

---

**Algorithm 3:** Zero-shot and Few-shot Q-learning with skill machines

**Input** : $\mathcal{S}, \mathcal{A}, \mathcal{P}, \mathcal{C}, \langle \mathcal{U}, u_0, \delta_u, \delta_Q \rangle, \gamma, \alpha$
**Initialise :** $Q(s, u, a)$

1 **foreach** *episode* **do**
2     Observe initial state $s \in \mathcal{S}$ and propositions $l \in 2^{\mathcal{P}}$, RM state $u \leftarrow u_0$ and constraints $c \leftarrow \emptyset$
3     **while** *episode is not done* **do**
4        **if** *zero-shot* **then**
5           $a \leftarrow \arg\max\limits_{a} \delta_Q(u)(\langle s, c \rangle, \langle a, 0 \rangle)$
6        **else**
          /* Fewshot by using $\delta_Q$ in the behaviour policy      */
7           $a \leftarrow$
$$\begin{cases} \arg\max\limits_{a} \left( \max\{\gamma Q(s, u, a), (1 - \gamma)\delta_Q(u)(\langle s, c \rangle, \langle a, 0 \rangle)\} \right) & \text{if } Bernoulli(1 - \epsilon) = 1 \\ \text{sample } a \in \mathcal{A} & \text{otherwise} \end{cases}$$
8     Take action $a$ and observe the next state $s'$ and true propositions $l'$
9     Get reward $r \leftarrow \delta_r(u)(s, a, s')$, true constraints $c \leftarrow c \cup ((\widehat{l} \oplus \widehat{l'}) \cap \mathcal{C})$,
10     and the next RM state $u' \leftarrow \delta_u(u, l')$
11     **if** $u \neq u'$ **then** $c \leftarrow \emptyset$
12     **if** $s'$ *or* $u'$ *is terminal* **then**
13        $Q(s, u, a) \xleftarrow{\alpha} r$
14     **else**
15        $Q(s, u, a) \xleftarrow{\alpha} \left[ r + \gamma \max\limits_{a'} Q(s', u', a') \right]$
16     $s \leftarrow s'$ and $u \leftarrow u'$

---

### A.3 DETAILS OF EXPERIMENTAL SETTINGS

In this section we elaborate further on the hyper-parameters for the various experiments in Section 4. We also describe the pretraining of WVFs for all of the experimental settings which corresponds to learning the task primitives for each domain. The same hyper-parameters are used for all algorithms in a particular experiment. This is to ensure that we evaluate the relative performance fairly and consistently. The full list of hyper-parameters for the Office World, Moving Targets and SafeAI Gym domain experiments are shown in Tables A1-A3 respectively.

| Hyper-parameter | Value |
|---|---|
| Timesteps | $1e^5$ |
| Training exploration ($\epsilon$) | 0.5 |
| Per-episode evaluation exploration ($\epsilon$) | 0.1 |
| Discount Factor ($\gamma$) | 0.9 |

Table A1: Table of hyper-parameters used for Q-learning in the Office World experiments.

To use skill machines we require pre-trained WVFs. As mentioned above, all WVFs are trained using the same hyper-parameters as any other training. Additionally, for all experiments the WVFs are pre-trained on the base task primitives for the domain. For example, in the Office World domain, the WVFs are trained on the $|\mathcal{P} \cup \mathcal{C}|$ base task primitives corresponding to achieving each predicate, $\mathcal{P} = \{A, B, C, D, \maltese, \text{⚒}, \boxtimes, \text{⛏}, \boxtimes^+, \text{⛏}^+\}$ (reaching states the predicate is set to True), with constraints $\mathcal{C} = \{\widehat{\maltese}\}$. All other primitives in this domain can be obtained zero-shot through value function composition. Similarly, for the moving targets domain (Figure A1), the WVFs are pre-trained on the primitives corresponding to obtaining objects by shape or colour in the environment separately, $\mathcal{P} = \{\square, \blacksquare, \blacksquare, \boxtimes, \bigcirc\}$, with constraints $\mathcal{C} = \widehat{\mathcal{P}}$. From here the value functions for finding objects of particular colours or any more complex primitives can be composed zero-shot. Finally, for the SafeAI Gym environment the base skill primitives correspond to going to a *cylinder* (🔴), a *button* (🟢), and a *blue region* (🔵): $\mathcal{P} = \{🔴, 🟢, 🔵\}$, trained with constraints $\mathcal{C} = \{\widehat{🔵}\}$.

| Hyper-parameter | Value |
| --- | --- |
| Timesteps | $1e^6$ |
| Neural Network architecture | $CNN + MLP$ |
| CNN architecture | Defaults of Mnih et al. (2015) |
| MLP hidden layers | $1024 \times 1024 \times 1024$ |
| Start exploration ($\epsilon$) | 1 |
| End exploration ($\epsilon$) | 0.1 |
| Exploration decay duration ($\epsilon$) | $5e^5$ |
| Discount Factor ($\gamma$) | 0.99 |
| Others | Defaults of Mnih et al. (2015) |

Table A2: Table of hyper-parameters used for Deep Q-learning in the Moving Targets experiments.

| Hyper-parameter | Value |
| --- | --- |
| Timesteps | $1e^6$ |
| Neural Network architecture | $MLP$ |
| MLP hidden layers | $2024 \times 2024 \times 2024$ |
| Max episodes length | 100 |
| Target noise | 0.2 |
| Action noise | 0.2 |
| Discount Factor ($\gamma$) | 0.99 |
| Others | Defaults of Achiam (2018) |

Table A3: Table of hyper-parameters used for the TD3 in the SafeAI Gym experiments.

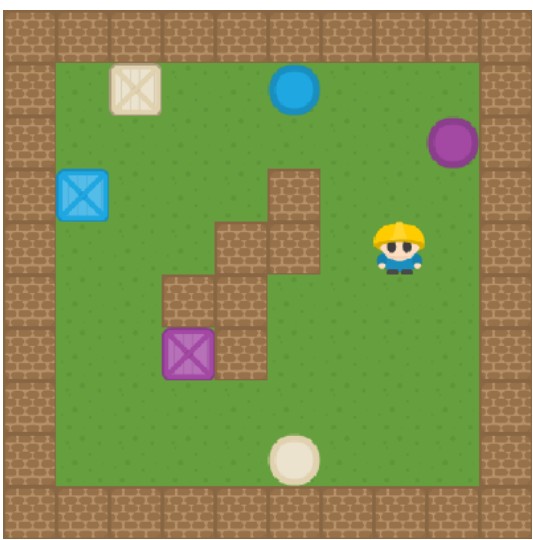

Figure A1: Moving Targets domain

## A.4 OFFICE GRIDWORLD ADDITIONAL EXPERIMENTS AND FIGURES

We illustrate the environment and an example temporal logic task in it in Figure A2. In this environment, an agent (blue circle) can move to adjacent cells in any of the cardinal directions ($|\mathcal{A}| = 4$) and observe its $(x, y)$ position ($|\mathcal{S}| = 120$). Cells marked ☕, ✉, and 🧍 respectively represent the coffee, mail, and office locations. Those marked � indicate decorations that are broken if the agent collides with them, and $A–D$ indicate the corner rooms. The reward machines that specify tasks in this environment are defined over 10 propositions: $\mathcal{P} = \{A, B, C, D, ✻, ☕, ✉, 🧍, ✉^+, 🧍^+\}$, where the first 8 propositions are true when the agent is at their respective locations, $✉^+$ is true when the agent is at ✉ and there is mail to be collected, and 🧍$^+$ is true when the agent is at 🧍 and there is someone in the office.

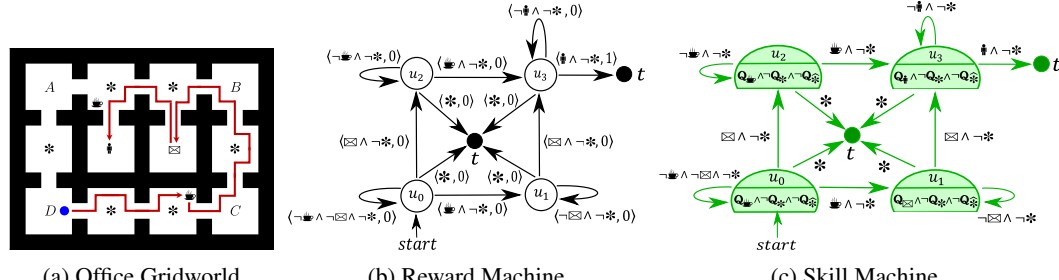

(a) Office Gridworld    (b) Reward Machine    (c) Skill Machine

Figure A2: Illustration of (a) the office gridworld where the blue circle represents the agent; (b) the reward machine for the task "deliver coffee and mail to the office without breaking any decoration", given by the LTL specification $\left(\left(F\left(☕ \wedge X\left(F\left(✉ \wedge X\left(F🚹\right)\right)\right)\right)\right) \vee \left(F\left(✉ \wedge X\left(F\left(☕ \wedge X\left(F🚹\right)\right)\right)\right)\right)\right) \wedge (G\neg✲)$; (c) the skill machine obtained from the reward machine which can then be used to achieve the task specification zero-shot—the red trajectory in (a). The nodes labeled $t$ represent terminal states.

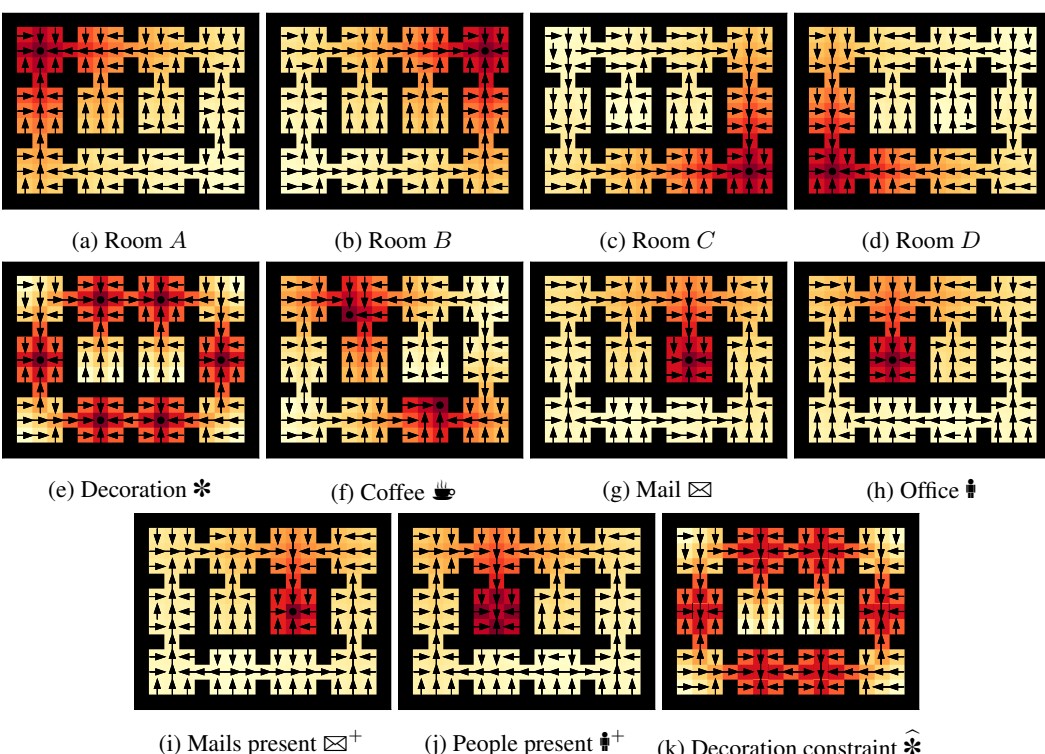

(a) Room $A$    (b) Room $B$    (c) Room $C$    (d) Room $D$

(e) Decoration ✲    (f) Coffee ☕    (g) Mail ✉    (h) Office 🚹

(i) Mails present $✉^+$    (j) People present $🚹^+$    (k) Decoration constraint $\widehat{✲}$

Figure A3: The policies (arrows) and value functions (heat map) of the primitive tasks in the Office Gridworld. These are obtained by maximising over the goals of the learned WVFs.

Figure A3 shows the skill primitives learned for each proposition in the environment, and Figure A4 shows the trajectories of our zero-shot agent (SM) and few-shot agent (QL-SM) for various tasks. Finally, we run 2 experiments to demonstrate the performance of our zero-shot and few-shot approach when the global reachability assumption does not hold.

1. **When the reachability assumption is not satisfied in some initial states:** In the first experiment (Figure A5), the agent needs to solve task 1 of Table 1 (($F(☕ \wedge X(F 🚹)))) \wedge (G \neg✲)$), but we modify the environment such that one of the coffee locations is absorbing (a sink environment state). This breaks the global reachability assumption since the agent can no longer reach the office location after it reaches the absorbing coffee location. As a result, we observe that the zero-shot agent (SM) is even more sub-optimal than before

because it cannot satisfy the task when it starts at locations that are closer to the absorbing coffee location. However, we can observe that the few-shot agent (QL-SM) is still able to learn the optimal policy, starting with the same performance as the zero-shot agent. Note that the hierarchical agent (HRM) also converges to the same performance as our zero-shot agent because it also tries to reach the nearest coffee location.

2. **When the reachability assumption is not satisfied in all initial states:** In the second experiment (Figure A6), the agent needs to solve the task with LTL specification $(F\ \text{👤})\wedge(\neg\text{👤}\ U\ \text{☕})$—the environment is still modified such that one of the coffee locations is absorbing. Here, the Boolean expression $\text{☕}\wedge\text{👤}$ is not satisfiable since there is no state where both propositions ($\text{☕}$ and $\text{👤}$) are true. Hence, this can be seen as the worst-case scenario for our approach (without outright making the task unsatisfiable), since $\text{☕}\wedge\text{👤}$ is the Boolean expression greedily selected in the starting RM state. As a result, our zero-shot agent completely fails to solve this task. Even in this case, we can observe that the few-shot agent is still able to learn the optimal policy.

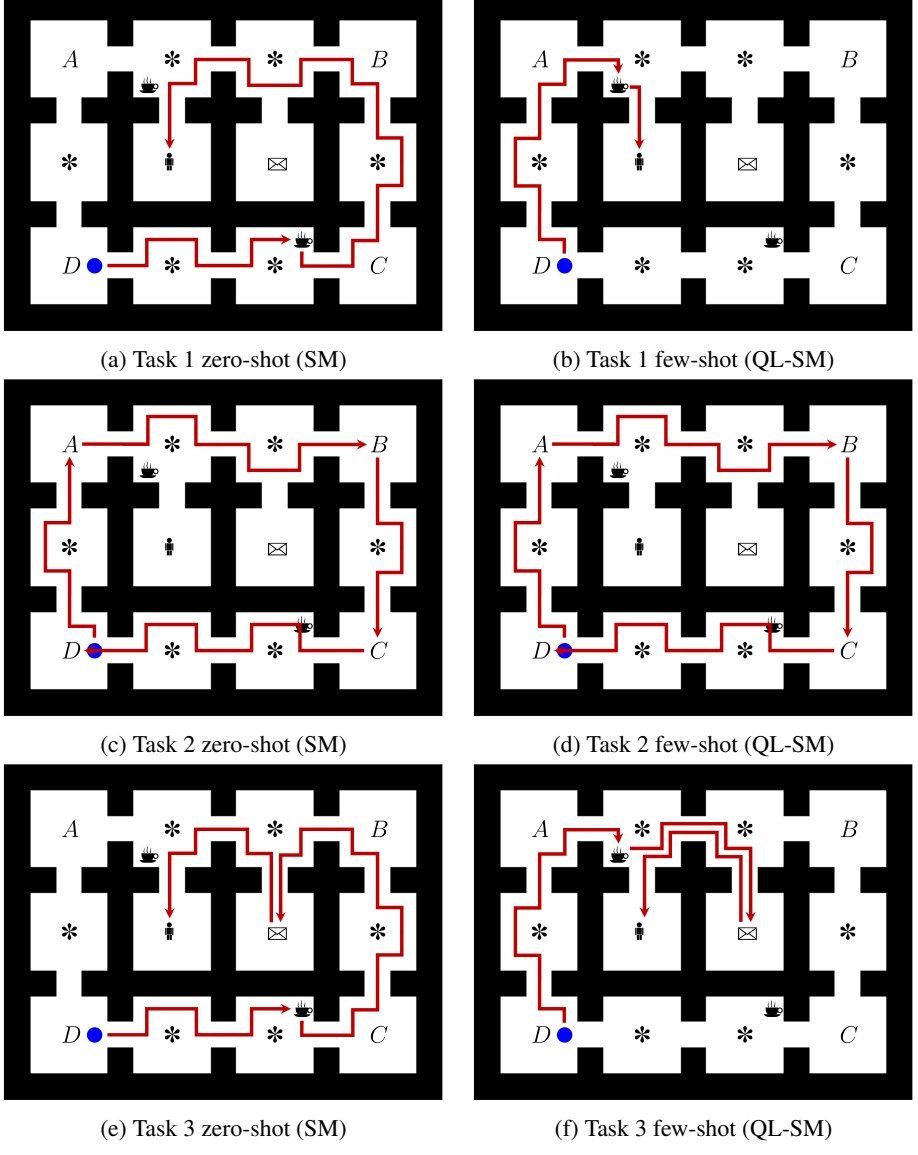

(a) Task 1 zero-shot (SM)  (b) Task 1 few-shot (QL-SM)

(c) Task 2 zero-shot (SM)  (d) Task 2 few-shot (QL-SM)

(e) Task 3 zero-shot (SM)  (f) Task 3 few-shot (QL-SM)

Figure A4: Agent trajectories for various tasks in the Office Gridworld (Table 1) using the skill machine without further learning (left) and with further learning (right).

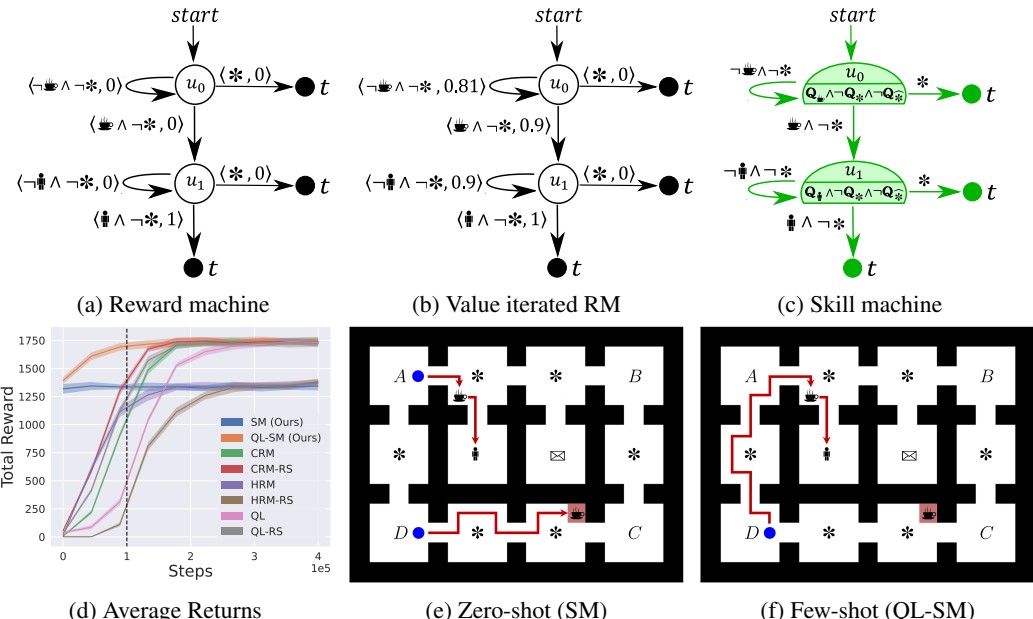

Figure A5: Results for the task with LTL specification $\left(F(\text{☕} \land X(F \text{🧍}))\right) \land (G \neg \text{✳})$ when the global reachability assumption does not hold. Here, the Office Gridworld is modified such that the position of the lower right coffee (the red cell) is made absorbing (the agent can not leave that state after reaching it). We show: (a) the reward machine for the task, (b) the value iterated reward machine (using $\gamma = 0.9$), (c) the resulting skill machine, (d) the resulting average returns compared to the baselines, (e) sample trajectories of the zero-shot agent, and (f) sample trajectory of the few-shot agent.

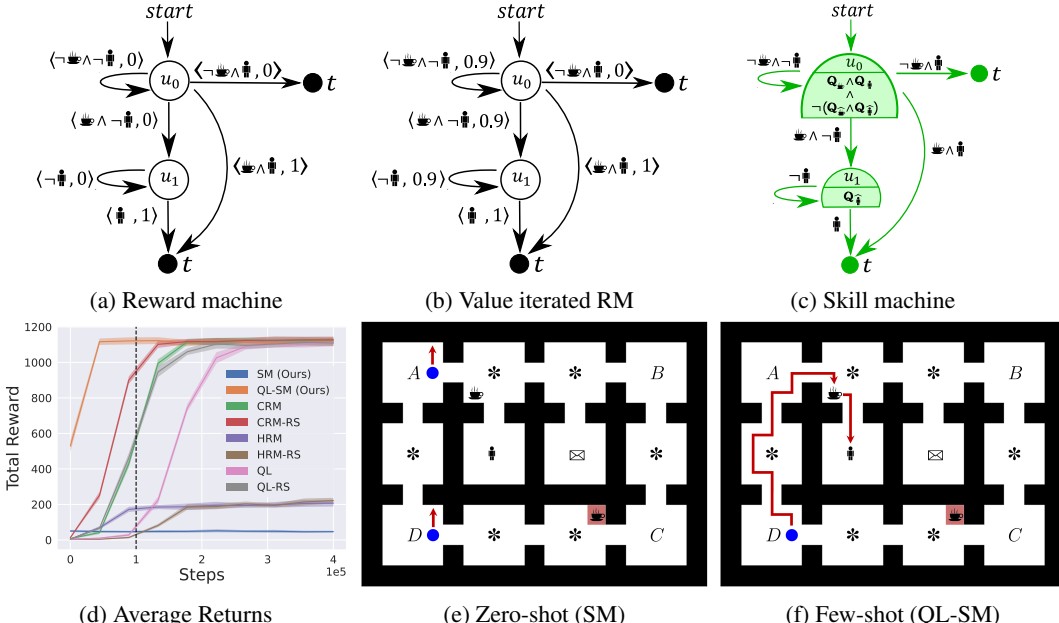

Figure A6: Results for the task with LTL specification $(F \text{🧍}) \land (\neg \text{🧍} \, U \, \text{☕})$ where the global reachability assumption is not satisfied. Here, the Office Gridworld is modified such that the position of the lower right coffee (the red cell) is made absorbing. We show (a) the RM for the task, (b) the value iterated RM (using $\gamma = 0.9$), (c) the resulting SM, (d) the resulting average returns compared to the baselines, (e) sample trajectories of the zero-shot agent, and (f) sample trajectory of the few-shot agent.

### A.5 Function Approximation with Continuous Actions and States

We demonstrate our temporal logic composition approach in a Safety Gym domain (Ray et al., 2019) which has a continuous state space ($\mathcal{S} = \mathbb{R}^{60}$) and continuous action space ($\mathcal{A} = \mathbb{R}^2$). The agent here is a point mass that needs to navigate to various regions defined by 3 propositions ($\mathcal{P} = \{\blacksquare, \bullet, \bigcirc\}$) corresponding to its 3 lidar sensors for the *red cylinder* ($\blacksquare$), the *green buttons* ($\bullet$), and the *blue regions* ($\bigcirc$). The agent, 4 buttons and 2 blue regions are randomly placed on the plane. The cylinder is randomly placed on one of the buttons. We first learn the 4 base skill primitives corresponding to each predicate (with constraints $\mathcal{C} = \{\widehat{\bigcirc}\}$), with goal-oriented Q-learning Nangue Tasse et al. (2020) but using Twin Delayed DDPG (Fujimoto et al., 2018) to update the WVFs. Figure A10 shows the trajectories of the SM policies for the six tasks described in Table A4. Our results shows that skill primitives can be leveraged to achieve zero-shot temporal logics even in continuous domains.

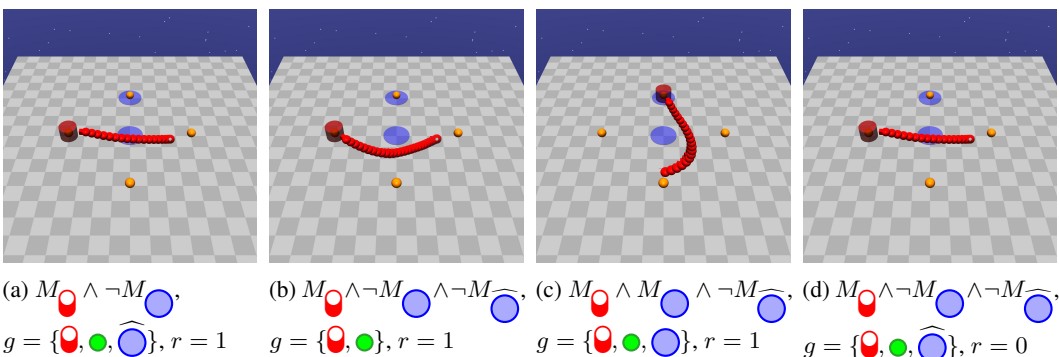

(a) $M_{\bigcirc} \wedge \neg M_{\bigcirc}$, $g = \{\blacksquare, \bullet, \widehat{\bigcirc}\}, r = 1$ (b) $M_{\blacksquare} \wedge \neg M_{\bigcirc} \wedge \neg M_{\widehat{\bigcirc}}$, $g = \{\blacksquare, \bullet\}, r = 1$ (c) $M_{\blacksquare} \wedge M_{\bigcirc} \wedge \neg M_{\widehat{\bigcirc}}$, $g = \{\blacksquare, \bullet, \bigcirc\}, r = 1$ (d) $M_{\blacksquare} \wedge \neg M_{\bigcirc} \wedge \neg M_{\widehat{\bigcirc}}$, $g = \{\blacksquare, \bullet, \widehat{\bigcirc}\}, r = 0$

Figure A7: Effect of constraints on primitives ($C = \{\widehat{\bigcirc}\}$). We show compositions of task primitives (for example $M_{\bigcirc} \wedge \neg M_{\blacksquare}$ where the agent needs to achieve $\blacksquare \wedge \neg \bigcirc$), trajectories, goal reached ($g$), and reward obtained ($r$) when following: (a-c) Optimal policies; and (d) a non-optimal policy.

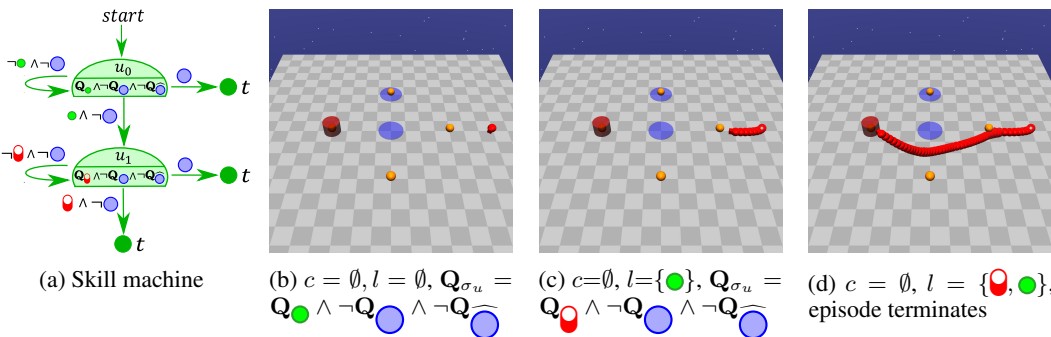

(a) Skill machine (b) $c = \emptyset, l = \emptyset, \mathbf{Q}_{\sigma_u} = \mathbf{Q}_{\bullet} \wedge \neg \mathbf{Q}_{\bigcirc} \wedge \neg \mathbf{Q}_{\widehat{\bigcirc}}$ (c) $c = \emptyset, l = \{\bullet\}, \mathbf{Q}_{\sigma_u} = \mathbf{Q}_{\blacksquare} \wedge \neg \mathbf{Q}_{\bigcirc} \wedge \neg \mathbf{Q}_{\widehat{\bigcirc}}$ (d) $c = \emptyset, l = \{\blacksquare, \bullet\}$, episode terminates

Figure A8: Execution of a skill machine in the Safety Gym domain. (a) An example skill machine; (b) A snapshot of the environment at the initial state. In this state, no constraint has been reached ($c = \emptyset$), no proposition is true ($l = \emptyset$), the SM is at state $u = u_0$, and the composed skill outputted by the SM is $\mathbf{Q}_{\sigma_u} = \mathbf{Q}_{\bullet} \wedge \neg \mathbf{Q}_{\bigcirc} \wedge \neg \mathbf{Q}_{\widehat{\bigcirc}}$ (which the agent uses to act in the environment); (c) The trajectory of the agent until it achieves $\bullet \wedge \neg \bigcirc \wedge \neg \widehat{\bigcirc}$. In the current environment state, no constraint has been reached ($c = \emptyset$), the agent is at a green button ($l = \{\bullet\}$), the SM transitions to state $u = u_1$, and the composed skill outputted by the SM is $\mathbf{Q}_{\sigma_u} = \mathbf{Q}_{\blacksquare} \wedge \neg \mathbf{Q}_{\bigcirc} \wedge \neg \mathbf{Q}_{\widehat{\bigcirc}}$ (which the agent uses to act in the environment); (d) The trajectory of the agent until the agent achieves $\blacksquare \wedge \neg \bigcirc \wedge \neg \widehat{\bigcirc}$. In the current environment state, no constraint has been reached ($c = \emptyset$), the agent is at the red cylinder and a green button ($l = \{\blacksquare, \bullet\}$), the SM transitions to the terminal state $t$, and the episode terminates.

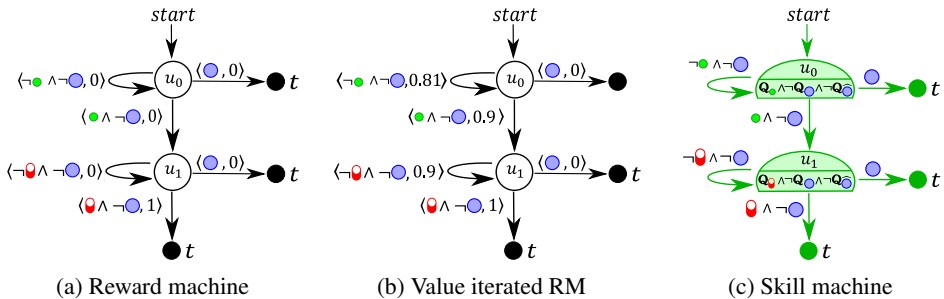

(a) Reward machine      (b) Value iterated RM      (c) Skill machine

Figure A9: The reward machine, value iterated reward machine (using $\gamma = 0.9$) and skill machine for the task with LTL specification $(F(\bullet \wedge X(F \bullet))) \wedge (G \neg \bigcirc)$. The agent composes its skill primitives to achieve $\sigma_{\mathcal{P}} \wedge \neg\sigma_{\mathcal{C}} = (\bullet \wedge \neg\bigcirc) \wedge \neg(\widehat{\bigcirc})$ at $u_0$ and $\sigma_{\mathcal{P}} \wedge \neg\sigma_{\mathcal{C}} = (\bullet \wedge \neg\bigcirc) \wedge \neg(\widehat{\bigcirc})$ at $u_1$.

| Task | Description — LTL |
|---|---|
| 1 | Navigate to a button and then to a cylinder. — $(F(\bullet \wedge X(F \bullet)))$ |
| 2 | Navigate to a button and then to a cylinder while never entering blue regions — $(F(\bullet \wedge X(F \bullet))) \wedge (G \neg\bigcirc)$ |
| 3 | Navigate to a button, then to a cylinder without entering blue regions, then to a button inside a blue region, and finally to a cylinder again. — $F(\bullet \wedge X(F((\bullet \wedge \neg\bigcirc) \wedge X(F((\bullet \wedge \bigcirc) \wedge X(F\bullet))))))$ |
| 4 | Navigate to a button and then to a cylinder in a blue region. — $(F(\bullet \wedge X(F \bullet \wedge \bigcirc)))$ |
| 5 | Navigate to a cylinder, then to a button in a blue region, and finally to a cylinder again. — $(F(\bullet \wedge X(F((\bullet \wedge \bigcirc) \wedge X(\bullet)))))$ |
| 6 | Navigate to a blue region, then to a button with a cylinder, and finally to a cylinder while avoiding blue regions. — $(F(\bigcirc \wedge X(F((\bullet \wedge \bullet) \wedge X((F \bullet) \wedge (G\neg\bigcirc))))))$ |

Table A4: Tasks in the Safety Gym domains. The RMs are generated from the LTL expressions.

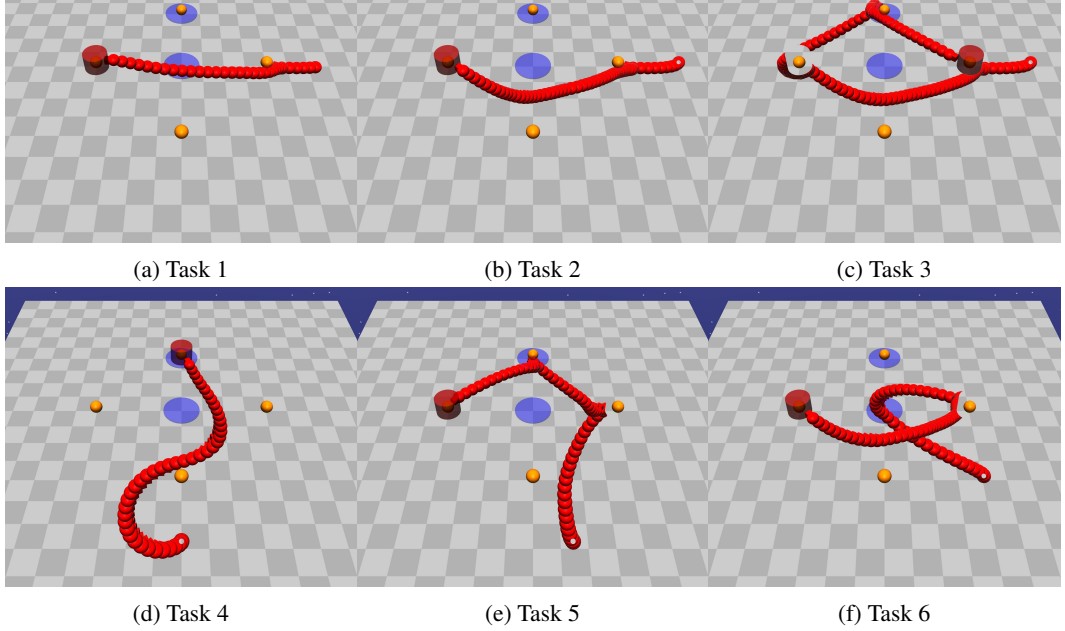

(a) Task 1      (b) Task 2      (c) Task 3

(d) Task 4      (e) Task 5      (f) Task 6

Figure A10: Visualisations of the trajectories obtained by following the zero-shot composed policies from the skill machine for tasks in Table A4.

