# OpenReview forum: "Skill Machines: Temporal Logic Skill Composition in Reinforcement Learning"
_ICLR.cc/2024/Conference — ICLR 2024 poster_

### Official Review · Reviewer_38Qb · 2023-10-30

**Soundness:** 2 fair
**Presentation:** 2 fair
**Contribution:** 2 fair
**Rating:** 5
**Confidence:** 5

**Summary:**

This work proposes skill machines to encode skill primitives of all high-level sub-tasks. Then zero-shot and few-shot learning is used to help train a policy which aims to satisfy specifications composing sub-tasks temporally and logically.

**Strengths:**

1. Proposed a principled way to transfer learned task policies to related temporal logic specification.
2. Conducted thorough experiments to validate the approach.

**Weaknesses:**

1. The definition of (linear) temporal logic is not provided.

2. I would suggest the authors to briefly mention how to convert LTL to reward machines, e.g., through buchi automata, to make it more clear.

3. This work only considers some simple temporal operations such as "until" and "eventually", but no continuous time interval. Extension to other temporal logic like signal temporal logic (STL) can be more interesting and useful practically -- also more challenging for sure.

4. There are some related work to compare. For instance, I find the following two papers relevant to this paper, the authors are encouraged to compare with them.
[1]. Luo, X. and Zavlanos, M.M., 2019, December. Transfer planning for temporal logic tasks. In 2019 IEEE 58th Conference on Decision and Control (CDC) (pp. 5306-5311). IEEE.
[2]. León, B.G., Shanahan, M. and Belardinelli, F., Systematic Generalisation of Temporal Tasks through Deep Reinforcement Learning.

**Questions:**

1. In Sec 2.2, the authors say "Consider the multitask setting, where all tasks share the same state and action space, transition
dynamics and discount factor, but differ in reward function. The agent is required to reach a set of desired terminal states in some goal space G ⊆ S." Does this mean all subtasks can be represented as reaching a some goal states? If yes, this should be remarked more clearly.

---

> ### Author Response · Authors · 2023-11-20
>
> We really appreciate the reviewer’s time and effort spent in reviewing our paper and providing useful feedback. We hope the reviewer’s concerns are fully addressed with our response here, and are happy to clarify any further ones during the discussion period.
>
> **W1:**
> - We have added the definition for LTL in Section 2.1 page 3. We thank the reviewer for this suggestion. Due to space constraints,  we refer the reader to Camacho et al. (2019) for an expanded explanation of LTL and its interaction with RL.
>
>
> **W2:**
> - Similarly, we have added a brief description of how to convert LTL to reward machines in the caption of Figure 1. Due to space constraints,  we again refer the reader to Camacho et al. (2019) for an expanded explanation of how to convert LTL to reward machines.
>
>
> **W3:**
> - We focus on LTL because of its relative simplicity and popularity in the literature, which is integral to not obfuscate the focus of this paper—which is about addressing the spatial and temporal curses of dimensionality by composing sub-tasks logically and temporally. Extending this work to other types of temporal logic like STL is definitely an interesting direction for future works. Since such an extension is not trivial, it is outside the scope of this work, but we believe that this work forms a good foundation for such extensions (since STL is a generalisation of LTL).
>
>
> **W4:**
> - We thank the reviewer for the suggested works, and we are happy to include them in our related works. We will further discuss them here and hope that this clarifies why we omitted them from our baselines.
>
>
> - To the best of our understanding, Luo et al. (2019) assume the usual planning/optimal-control setting, which is a very different setting to ours and all of the prior works cited in this paper. Mainly, they assume that the environment dynamics are known and amenable to planning. Here, they are more interested in how to efficiently obtain and transfer plans from previous LTL tasks to new ones. Hence, they also do not compare against reward machines but instead compare against a prior work on temporal logic optimal-control. Note that in our work and prior works, the environment dynamics are not known (hence why RL is used), hence we are unsure how the suggested baseline can be applied in this setting (since planning at the level of dynamics is not possible). In our work and prior works, planning (if used) only happens at the level of the reward machines.
>
> - We believe that CRM is representative of León et al. (2019) in our Figure 3 experiments. Since our work is the first to attempt and achieve both spatial and temporal skill composition in temporal logic tasks provably (to the best of our knowledge), the aim of this experiment was to demonstrate the importance and benefit of our approach compared to the other popular ones considered in prior works. To elaborate slightly:
>
>   -  In León et al. (2019), assuming that the tasks are specified using TTL, they propose using a combined or separate task and observation module (LSTMs and CNNs) to end-to-end learn an embedding of “TTL progressions”. They then demonstrate empirically that their approach can learn given tasks and also achieve some generalisation to new ones, with varying success depending on the architectures used. However, CRMs and León et al. (2019) do not leverage spatial or temporal skill composition. Hence, they rely on function approximation (e.g neural networks) to generalise both spatially and temporally. Hence, considering the spatial and temporal curses of dimensionality described in our paper (and the arbitrarily long temporal sequence of sub-tasks present in each task), they have to learn every new task with high probability. Consequently, CRMs provide a baseline representative of the class of algorithms without spatial or temporal composition that instead leverages the compositional structure of temporal logic to learn optimal policies efficiently.
>
>
> - Thus, we are uncertain what benefit a comparison with León et al. (2019) would provide which is not already achieved by comparing with CRMs, and would appreciate the reviewer's thoughts on this point. We do still agree that these are important citations and will gladly include them as such.
>
>
> **Q1:**
> - Yes. We thank the reviewer for the suggestion and have clarified this sentence in the background (Sec 2.2 Page 3).

---

> > ### Author Response · Authors · 2023-11-22
> > **Following Up with Reviewer 38Qb**
> >
> > We thank the reviewer once again for their time and consideration of our work.
> >
> > We would just like to follow up on whether there are any remaining questions or concerns that we can address as the discussion period draws to a close.

---

### Official Review · Reviewer_pDWg · 2023-10-31

**Soundness:** 3 good
**Presentation:** 2 fair
**Contribution:** 2 fair
**Rating:** 6
**Confidence:** 4

**Summary:**

The paper introduces skill machines, which are finite state machines derived from reward machines, enabling agents to tackle complex tasks involving temporal and spatial composition. These skills are encoded in a specialized goal-oriented value function. By combining these learned skills with the value functions, downstream tasks can be solved without additional learning. Importantly, this method guarantees that the resulting policy aligns with the logical task specification. The behavior generated is provably satisficing, with empirical evidence indicating performance close to optimality. The paper suggests that further fine-tuning can enhance sample efficiency if optimal performance is required.

**Strengths:**

The paper showcases the development of an agent with the ability to flexibly compose skills both logically and temporally to provably achieve specifications in the regular fragment of linear temporal logic (LTL). This approach empowers the agent to sequence and order goal-reaching skills to satisfy temporal logic specifications. Zero-shot generalization is important in this context is important due to the vastness of the task space associated with LTL objectives,  rendering training across every conceivable scenario intractable.

**Weaknesses:**

The paper raises concerns about the optimality of the policy resulting from composition, as the task planning algorithm lacks consideration for the cost of sub-tasks within a skill machine. Consequently, the generated behavior can be suboptimal due to this oversight. Although the paper presents a method for few-shot learning to create new skills, it remains unclear why learning a new skill is preferred over recomposing existing ones, raising questions about the approach's efficiency. Furthermore, it does not seem a learned new skill in few-shot learning is consistent with Definition 3.1.

An empirical comparison between the paper's few-shot learning strategy and the Logical Options Framework (LQF) baseline is lacking. The LQF baseline employs a meta-policy for choosing options to achieve subgoals within the finite state automaton representation of an LTL property. This approach integrates value iteration over the product of the finite automaton and the environment, ensuring that options can be recombined to fulfill new tasks without necessitating the learning of entirely new options.

Furthermore, the absence of a comparison with LTL2Action in the continuous safety gym environment raises questions. The paper concludes that the zero-shot agent's performance is nearly optimal. It is crucial to assess this claim by comparing the zero-shot agent's performance in terms of discounted rewards with that of the LTL2Action agent in the safety gym environment.

There is room for improvement in the execution of the paper, particularly in the clarity of its formalization. Following the formalization proved to be challenging, and below, I have outlined my specific concerns regarding this aspect.

Brandon Araki, Xiao Li, Kiran Vodrahalli, Jonathan A. DeCastro, Micah J. Fry, and Daniela Rus. The logical
options framework. CoRR, abs/2102.12571, 2021

**Questions:**

* In Sec 2.2, $\pi^\ast(s) \in arg max_a max_g \bar{Q^\ast}(s, g, a)$.  I would appreciate clarification regarding the reward function associated with the policy $\pi^\ast$. Does this imply that the policy is considered successful whenever it reaches any goal $g$ within the goal space?

* In Definition 3.1, what is the purpose of tracking $c$ as a history of safety violation to constraints in $\mathcal{C}$?

* In the work by Nangue Tasse et al. (2020), the reward function for a skill specifies that if the agent enters a terminal state corresponding to a different task, it should incur the most substantial penalty possible. I am curious why this aspect is omitted in Definition 3.1.

* At state $u$ in a skill machine, it computes a skill $\delta Q(u)(\langle s, c\rangle,\langle a, 0 \rangle) \mapsto max_g
\bar{Q}^\ast_u (\langle s, c \rangle, g, \langle a, 0 \rangle)$ that an agent can
use to take an action $a$.  Is it necessary for this skill to enumerate the entire goal space? At every state within the skill machine, you have already derived a Boolean combination of primitive Q functions concerning specific goals. For instance, in $u_0$ from Fig. 1, the relevant goals include the green button and the blue region. Could we potentially use the green button directly as the goal for Q_button and the blue region as the goal for Q_blue, thereby bypassing the need to exhaustively enumerate all possible goals?

* Relatedly, in the scenario where the goal space is continuous, how does the algorithm determine the optimal value for $g$?

* As per Definition 3.1, each skill is associated with a sub-goal. However, in the context of few-shot learning, what constitutes the goal for a newly acquired skill?

---

> ### Author Response · Authors · 2023-11-20
> **Official Comment by Authors (Part 1/2)**
>
> We really appreciate the reviewer’s time and effort spent in reviewing our paper and providing useful feedback. We hope the reviewer’s concerns are fully addressed with our response here, and are happy to clarify any further ones during the discussion period.
>
> > Weaknesses
>
> **W1: [Few-shot learning]**
> - Please note that we do not claim that our few-shot approach is the best way to improve our zero-shot policy. In fact, by using Q-learning to learn a new skill, our aim here was to use a relatively simple approach to not obfuscate the main point of this section (which is that our zero-shot policy can be improved in a way that is guaranteed to converge to the optimal policy).
>
> - Investigating other few-shot methods for improving our zero-shot policy is definitely an interesting direction for future works. For example, by designing efficient algorithms for learning the best compositions of skill primitives to use.
>
> - (also from Q6) In the few-shot learning case, the new skill learned is not a skill primitive, but just a regular action-value function for the specific task. We have clarified this point on page 6 (the highlighted text).
>
> **W2: [Baselines]** This is an important concern from the reviewer, and we hope our discussion here fully clarifies why we omitted the 2 suggested papers from our baselines (despite citing them).
>
> - We believe that HRM [Icarte et al., 2022] is representative of LOF [Araki et al., 2021] and that CRM [Icarte et al., 2022] is representative of LTL2Action [Vaezipoor et al., 2021] in our Figure 3 experiments. Since our work is the first to attempt and achieve both spatial and temporal skill composition in temporal logic tasks provably (to the best of our knowledge), the aim of this experiment was to demonstrate the importance and benefit of our approach compared to the other popular ones considered in prior works. To elaborate slightly:
>
>   - While approaches that only leverage temporal skill composition like HRMs and LOF are able to learn tasks fast, they all still require learning the sub-task options for most new tasks. They are also all hierarchically optimal, while our few-shot approach is globally optimal (Figure A5 and A6 shows how the performance of a hierarchically optimal policy can be vastly lower than that of a globally optimal policy).
>
>   - In LOF, the authors learn options for each subgoal proposition of the reward machine, then use value iteration or Q-learning to learn a meta-policy over options to solve the **current** task. These options can then be reused in new tasks (i.e new reward machines) where the same sub-goal propositions appear. As described in the introduction and observed by Liu et al. (2022), in all these works (similarly to HRMs), the options learned from a previous task can not be transferred satisfactorily to some new tasks (also see the example on page 4). For example in the office world, if in a previous task the agent learns an option for “getting coffee”, it can reuse it to “get coffee then deliver it to the office”, but it cannot reuse it in a new task where it needs to “get coffee without breaking decorations”. In contrast, Skill Machines (our work) are able to bypass this issue by learning skill primitives that can then be composed zero-shot to get coffee with or without constraints. Thus, HRMs and LOF omit spatial compositionality and HRMs provide an appropriate baseline which is representative of the limitations of all such algorithms without spatial compositionally that instead leverages temporal compositionally to learn hierarchically-optimal policies.
>
>   - In LTL2Action, assuming that the tasks are specified using LTL, they propose using a separate LTL module (a graph neural network) to end-to-end learn an embedding of “LTL progressions”. They then demonstrate empirically that their approach converges to more optimal policies than a myopic baseline and also achieves comparable or better generalisation. However, CRMs and LTL2Action do not leverage spatial or temporal skill composition. Hence, they rely on function approximation (e.g neural networks) to generalise both spatially and temporally. Hence, considering the spatial and temporal curses of dimensionality described in our paper (and the arbitrarily long temporal sequence of sub-tasks present in each task), they have to learn every new task with high probability. Consequently, and similarly to the relationship between HRMs and LOF, CRMs provide a baseline representative of the class of algorithms without spatial or temporal composition that instead leverages the compositional structure of temporal logic to learn optimal policies efficiently.
>
> - Thus, we are uncertain what benefit a comparison with LOF or LTL2Action would provide which is not already achieved by comparing with HRMs and CRMs, and we would appreciate the reviewer's thoughts on this point. We do still agree that these are relevant citations, reflected by the fact that we did cite them in the original version.

---

> > ### Author Response · Authors · 2023-11-20
> > **Official Comment by Authors (Part 2/2)**
> >
> > > Questions
> >
> > **Q1:**
> > - In Sec 2.2, $\pi^*$ is the optimal policy for the task with reward function $R(s,a)$. Hence $\pi^*(s) \in argmax_a max_g \bar Q^*(s,g,a)$ just means that maximising over goals and actions (that is $argmax_a max_g \bar Q^*(s,g,a)$) is optimal with respect to the original task reward function $R(s,a)$. We have added a task label ($M$) to the policies and value functions to improve clarity on this point.
> >
> > **Q2:**
> > - It gives a Markov way for an agent to learn an optimal policy that does not violate safety constraints while trying to achieve sub-tasks. By making the history of safety violations reflected in the agent’s observations and the goals achieved in a sub-task, we just need to look at the goal achieved (which includes the safety violations) to reward the agent appropriately. We have added Figure A7 to illustrate this.
> >
> > **Q3:**
> > - The reward function $R_p$ in Definition 3.1 is not an “extended reward function” ($\bar R_p$). It is the regular reward function of a task primitive. The corresponding skill primitive $\bar Q_{p}^{*}$ (which is a WVF) indeed uses $\bar R_p$, which extends the regular reward function ($R_p$) of the task primitive to include the penalty $R_{MISS}$. Since our rewards are only $0$ or $1$, we use $R_{MISS}=R_{MIN}=0$ (following [Nangue Tasse et al., 2022])
> >
> > **Q4:**
> > - Yes, we can use the derived Boolean expression to reduce the number of goals to enumerate, but not always. For example, in $u_0$ from Fig. 1, we know (from the Boolean expression) that the max goal shouldn’t include “blue region”. So we can reduce the number by only enumerating all the goals that do not include “blue region”.
> >
> > - However, we can not just use {“green button”} directly as the goal for $Q_{button}$ and {“blue region”} as the goal for $Q_{blue}$, for multiple reasons. One reason is that the compositions are pointwise ($\[Q_{button} \wedge Q_{blue}\](s,g,a) = min${$Q_{button}(s,g,a), Q_{blue}(s,g,a) $}), so we cannot use different goals for $Q_{button}$ and $Q_{blue}$  inside the $min$. Also, {“green button”} and {“green button”,”red cylinder”} are both good goals for $Q_{button}$ , but we do not know which one is closest to the agent.
> >
> > **Q5:**
> > - The goal space $\mathcal{G} = 2^\mathcal{P}$ is always finite, since $\mathcal{P}$ is finite.
> >
> > **Q6:**
> > - In the few-shot learning case, the new skill learned is not a skill primitive, but just a regular action-value function for the specific task. We have clarified this point on page 6 (the highlighted text).

---

> > ### Comment · Reviewer_pDWg · 2023-11-22
> > **Comparison with LOF**
> >
> > I believe there is a misunderstanding about LOF. LOF assumes safety propositions in an RL environment with LTL objectives, learning options to reach subgoals while ensuring safety propositions are never met by the options. It then learns a meta-policy via value iteration to choose options for achieving subgoals within the finite state automaton representation of an LTL property. Thus, LOF can combine options to fulfill new tasks without needing to learn new options. I disagree with the statements that "LOF omits spatial compositionality".
> >
> > In this manuscript, safety constraints are similarly introduced, and I don't see the setup of safety constraints versus safety propositions as different. The proposed approach also uses value iteration to learn to combine options (or primitives, as the authors term them) but requires few-shot learning of a new "skill" to ensure optimality.
> >
> > The key difference, in my opinion, is that during value iteration, LOF considers the performance of an option to choose the best combination, whereas the proposed approach does not consider the reward performance of the primitives in this process, necessitating the learning of a new "skill" to compensate. This is why I consider LOF a relevant baseline for understanding the tradeoffs of these different approaches.
> >
> > Regarding the comparison with LTL2Action, I agree with the authors that this approach differs significantly. However, I suggest considering LTL2Action as an upper bound on policy performance. If the authors claim that their few-shot learning approach achieves global optimality, it would be beneficial to see if their reward performance indeed matches or is close to that of LTL2Action.
> >
> > Based on these points, my stance remains on the negative side.
> >
> > [LOF] Brandon Araki, Xiao Li, Kiran Vodrahalli, Jonathan A. DeCastro, Micah J. Fry, and Daniela Rus. The logical options framework. ICML 21.

---

> > > ### Comment · Reviewer_uyfa · 2023-11-22
> > > **In Defense of Author's LOF claims**
> > >
> > > I wish to clarify the LOF claims of the authors hopefully with some examples. Reviewer pDWg is indeed correct in stating that LOF can handle the avoidance tasks through the use of safety propositions by ensuring that all propositional policies avoid satisfying all safety propositions throughout their execution, thus demonstrating avoidance behavior. However the author's claims are slightly different here:
> > >
> > > 1) To demonstrate the avoidance behavior, the system designers must include $decorations$ as a part of the safety propositions in LOF, otherwise LOF policies will not prevent avoidance behavior at execution time. Also as a side effect the safety proposition avoidance will be seen for all downstream tasks if that policy is used. Therefore if an avoidance behavior is not necessary, then LOF must be pre-trained with options to turn avoidance behaviors on and off
> > > 2) Thus for 1 sub-task proposition, and $n$ safety propositions, the LOF library must train $2^n$ options for each avoidance prosposition. However skill machines Q-function composition will require training only $n+1$ options and initializing the Q-function for any requisite avoidance through logical composition.
> > > 3) This is an important capability difference between LOF and skill machines. This is also a difference that may not be uncovered simply through a comparative evaluation and comparing the task completion rates. The difference in behavior will only be uncovered if the task environment is configured such that in satisfying the reach proposition, the safety proposition will be along the way.
> > > 4) The author's argument is not that LOF cannot demonstrate avoidance behaviors, but rather it can demonstrate avoidance behaviors without explicitly training for it through logical composition approach presented by the authors.
> > >
> > > Thus I would argue that the there is enough differences between LOF, LTL2Action and skill machines. I also believe that qualitatively skill machines handles tasks that LOF cannot, and while I think the reverse is true as well, this is not a disqualifying criterion for skill machines to be accepted for publication.

---

> ### Comment · Reviewer_pDWg · 2023-11-22
> **Thanks for the clarification. Still unconvinced.**
>
> **1. To demonstrate the avoidance behavior, the system designers must include as a part of the safety propositions in LOF, otherwise LOF policies will not prevent avoidance behavior at execution time.**
>
> Indeed, LOF does require safety propositions in advance. However, skill machines also need predefined safety constraints to learn the skill primitives (as per Definition 3.1).
>
> **2. Thus for 1 sub-task proposition, and $n$ safety propositions, the LOF library must train $2^n$ options for each avoidance prosposition.**
>
> According to "Algorithm 1 Learning and Planning with Logical Options" in the LOF paper, it only needs to learn one option per sub-task proposition, regardless of the number of safety propositions.
>
> **3. The difference in behavior will only be uncovered if the task environment is configured such that in satisfying the reach proposition, the safety proposition will be along the way.**
>
> This is precisely why I find the LTL2Action environments to be valuable.
>
> **4. SKill machine can demonstrate avoidance behaviors without explicitly training for it through logical composition**
>
> I agree demonstrating avoidance behaviors without explicitly training would be a valuable feature. However, I have reservations about whether skill machines indeed have this feature. From the paper, before Definition 3.1, "By setting the blue region proposition as a constraint, the agent keeps track (in its cross-product state) of whether or not it has reached a blue region in its trajectory when learning a primitive". It seems that this aligns exactly with what LOF does for learning options to avoid safety propositions.

---

> > ### Comment · Reviewer_uyfa · 2023-11-22
> > **Contd.**
> >
> > Thank you for the clarification. I agree that LTL2Action environments are valuable, and that LTL2Action is an applicable baseline but do not believe that it would be a disqualifying factor simply because of how distinct these approaches are. While we disagree on whether this might be a disqualifying weakness, I think we agree in principle.
> >
> > On LOF:
> >
> > 1) I think the paper indeed makes the claim that compositional avoidance behavior can be achieved without explicitly training for it, case in point the moving targets domain where the set of subgoals and the set of constraints is identical (Section.
> > 2) In contrast in the LOF paper, the statement of Algorithm 1 holds only for a given fully defined safety automaton. The authors further clarify this in Section (A.3) (https://arxiv.org/pdf/2102.12571.pdf), and I quote the authors here:
> > > Option policies are learned by training on the product of the environment and the safety automaton, $\mathcal{E}×\mathcal{W}_{safety}$
> > and
> > > This is because in LOF, safety properties are not composable, so using a learning algorithm that is satisfying and optimal but not composable to learn the safety property is appropriate.
> >
> > Finally, the authors also mention that the safety property must not change throughout the task (depending on the liveness state), whereas skill machines places no such restrictions. Again, from the LOF authors:
> >
> > >Note that since the options are trained independently, one limitation of our formulation is that the safety properties cannot depend on the liveness state. In other words, when an agent reaches a new subgoal, the safety property cannot change. However, the workaround for this is not too complicated. First, if the liveness state affects the safety property,
> > this implies that liveness propositions such as subgoals may be in the safety property. In this case, as we described above, the subgoals present in the safety property need to be substituted with “safety twin” propositions. Then during option training, a policy-learning algorithm must be chosen that will learn sub-policies for all of the safety property states.
> >
> > Note that the work around that the authors suggest is specifically learning a different property for all possible safety constraints (or safety automata) that the agent would encounter during test execution. At face value it may seem that defining constraints the way the authors have done here is similar to the way LOF authors define safety propositions, however the key difference here is that the LOF authors require safety and subgoal propositions to be mutually exclusive sets with some work arounds proposed, whereas skill machines does not require this assumption to hold.
> >
> > 3) On the other hand Paragraph 2 page 4 explicitly motivates the need to compose avoidance constraints on the fly, and the evaluations in the moving target domain are geared towards test-time safety constraint composition. Which leads me to believe that skill machines is capable of such compositional behavior.
> >
> > Hopefully I have successfully conveyed my understanding of the way LOF and skill machines define avoidance behaviors, and please feel free to point out any gaps.

---

> > > ### Comment · Reviewer_pDWg · 2023-11-22
> > >
> > > Thank you. I now realize that the ability for compositional avoidance is an advantage of skill machines over LOF. My initial concern was focused on learning a new skill versus learning a new combination of existing skills, as in the LOF approach. The discussion on compositional avoidance might be slightly off the intended track. My concern remains after this discussion. However, it is good to know that this paper contributes to the advancement of existing methods, at least in one aspect. Based on this, I increased the score to 6.

---

> > > > ### Author Response · Authors · 2023-11-23
> > > >
> > > > We want to thank reviewer pDWg for their time and consideration of our work. We really appreciate the effort they have spent engaging during this discussion period, and are glad they have a better understanding of our work and contributions.
> > > >
> > > > We are in full agreement with the succinct clarifications provided by reviewer uyfa, and will certainly ensure that the prior approaches raised during this discussion period are discussed in Section 4 where we describe our experimental design and methodology. This will include the point of how these similar baselines relate to the benchmarks which we do use in our experiments.
> > > >
> > > > Finally, we understand reviewer's pDWg desire to still see a comparison with LOF in terms of few-shot performance. We have been familiarising ourselves with their code base (https://github.com/braraki/logical-options-framework), and have been able to implement the office grid world domain and tasks in it (the tasks for Figures 2, A5, and A6). We chose the office grid world since their provided code base is for the tabular setting. While the experiments are still running, our preliminary tests indicate that LOF performs similarly to HRM. Precisely, our discussion around Figure 2 remains unchanged, and LOF converges to the same performance as HRM in Figure A5 (this is because they are both hierarchically optimal). We commit to adding the results from these experiments to the subsequent revision once they finish running. We hope this will further increase the reviewer's confidence in our contributions.

---

### Official Review · Reviewer_vdA3 · 2023-11-01

**Soundness:** 3 good
**Presentation:** 2 fair
**Contribution:** 3 good
**Rating:** 6
**Confidence:** 3

**Summary:**

This paper points out that a large number of combinations of high-level goals can lead to the curse of dimensionality. To address this issue, this paper presents skill machines, which flexibly compose a set of skill primitives both logically and temporally to obtain near-optimal behaviors. This paper demonstrates that skill machines can map complex temporal logic task specifications to near-optimal behaviors, which are sufficient for solving subsequent tasks without further learning.

**Strengths:**

The motivation for this paper is quite sound. The agent begins by converting the LTL task specification into a reward machine (RM) and proceeds to determine the suitable spatial skill for each temporal node through value iteration. It also composes its skill primitives into spatial skills, effectively forming a skill machine. Finally, the agent applies these skills to solve the task without requiring additional learning.

**Weaknesses:**

One weakness of the paper lies in its relatively complex non-end-to-end training process. More precisely, it necessitates reinforcement learning training for acquiring skill primitives and additional value iteration for the selection of appropriate spatial skills. Another potential limitation is its applicability, which may be more suited to navigation scenarios. It requires different linear temporal logic (LTL) task specifications tailored to specific application scenarios, potentially leading to challenges in generalization across various scenarios. What’s more, the descriptions of definitions 3.1 and 3.2 are not clear enough and lack some corresponding explanations.

**Questions:**

1. How are different tasks specifically set in each benchmark? Why does the agent achieve zero-shot performance?

2. The policy generated by Skill Machine (SM) may be locally optimal, so Theorem 3.4 assumes global reachability and the policy is satisficing. There may be enough combinations of skills to approach this assumption, but it will limit the agent's exploration ability. And what is the approximate performance gap when certain states are unreachable?

---

> ### Author Response · Authors · 2023-11-20
>
> We really appreciate the reviewer’s time and effort spent in reviewing our paper and providing useful feedback. We hope the reviewer’s concerns are fully addressed with our response here, and are happy to clarify any further ones during the discussion period.
>
> > Weaknesses
>
> **W1:**
> - Please note that the use of RL to acquire skills is standard in the literature. Hence, while the need for RL to learn the skill primitives is indeed a weakness of this work, it is also a weakness of all prior works that do not assume known environment dynamics (e.g [Vaezipoor et al., 2021; Jothimurugan et al., 2021; Liu et al., 2022; Icarte et al. 2022]).
>
> - However, investigating approaches for acquiring skill primitives without RL is still an interesting direction for future works. For example, by using planning when the environment dynamics are known or behaviour cloning/imitation learning/offline RL when appropriate trajectories datasets are available.
>
> - Please note that the planning (value iteration) done in this work is only over the reward machines (it does not include the environment dynamics), and hence uses no additional environment samples. Value iteration is also a standard algorithm in RL [Sutton & Barto, 2009].
>
> **W2:**
> - The focus on LTL (and regular languages in general) indeed limits the applicability of this work, and to the best of our knowledge, similarly limits the applicability of all prior works in this litterature (e.g [Vaezipoor et al., 2021; Jothimurugan et al., 2021; Liu et al., 2022; Icarte et al. 2022]).
>
> - There is however strong literature justifying the benefit of specifying tasks using LTL (and formal languages in general) rather than scalar rewards (e.g [Li et al., 2017; Littman et al., 2017]). The main ones are the ease of specifying desired/undesired behaviours and sample efficiency.
>
> - We use LTL for navigation tasks in our experiments since they are a canonical scenario used in the literature, and are easier to understand (e.g [Vaezipoor et al., 2021; Jothimurugan et al., 2021; Liu et al., 2022; Icarte et al. 2022]). LTL can also be used to specify other types of tasks, like robot reach&manipulation tasks (e.g [Li et al., 2017; Jothimurugan et al., 2021; Araki et al., 2021;])
>
> **W4:**
> - We have added Figures A7 and Figure A8 to further improve the understanding of Definitions 3.1 and Definition 3.2. We hope these help clarify the reviewer’s concern here, and are happy to further clarify any aspect of these definitions.
>
>
> >  Questions
>
> **Q1:**
> - For each task in each benchmark, the RM is obtained by converting the LTL expression into an FSM using Duret-Lutz et al. (2016) (https://spot.lre.epita.fr/app/), and then giving a reward of 1 for accepting transitions and 0 otherwise. We then use the code base of Icarte et al. (2022) (https://github.com/RodrigoToroIcarte/reward_machines) to set the tasks for each RM.
>
> - Our agent achieves zero-shot performance because at each FSM state, the learned skill primitives can be composed to achieve the Boolean expression obtained from value iteration (if it is achievable). We have added Figure A8 to make this process clearer.
>
> **Q2:**
> - We have added two experiments (Figure A5 and Figure A6 on page 18) where we evaluate the performance of our approach in different cases when the reachability assumption does not hold. Please see the highlighted text in Section A.4 for the discussion. In brief, we observe that the performance of the zero-shot agent may significantly decrease when the reachability assumption does not hold in some states (Figure A5), and may lead to complete failures in the worst-case scenario when the reachability assumption does not hold in all states (Figure A6). In both cases, we observe that the few-shot agent can still quickly improve on the zero-shot performance by learning the optimal policy.

---

> > ### Comment · Reviewer_vdA3 · 2023-11-22
> > **Response**
> >
> > Dear authors,
> >
> > Thank you for providing the response to address my questions, which has partially alleviated my concerns. While the authors have cited several prior studies tested on manipulation tasks, it appears that this paper itself has not conducted such evaluations. Additionally, the issue of generalization still seems to be unresolved. I will temporarily withhold my score, but I also believe that this paper has a certain value and its quality is close to the acceptance standards of ICLR. If this paper is accepted, I will not hold any objections. Meanwhile, I will refer to the opinions of other reviewers and maintain consistency with the majority of them.

---

> > > ### Author Response · Authors · 2023-11-22
> > >
> > > Thank you to the reviewer for checking our response. We are glad that it has clarified some of the reviewer's concerns. Regarding the outstanding ones:
> > >
> > > **Types of tasks:**
> > > - We thought the reviewer's concern here was about the applicability of LTL (or temporal logic specified tasks in general), and that it may be more suited to navigation scenarios. Our citations were to address that concern. We apologise if we misunderstood the reviewer's concern here.
> > >
> > > - Please note how the semantic nature of tasks such as navigation, arm reach, or object manipulation is not of relevance in this work and prior works. The agent does not know this semantic nature, and there is no assumption in this work and prior works that make use of that semantic nature. The agent only assumes given sensors to detect when propositions are achieved (i.e a labeling function). We have treated environment observations in our theory as coming from an arbitrary environment. Hence, the types of tasks we considered in our experiments were specifically chosen to demonstrate the generalisation of our approach across various representative practical scenarios:
> > >   - Across different LTL specifications. For example, the LTL tasks in Table 1 for the office grid world, Table 2 for the moving targets domain, and Table A4 for safety gym.
> > >   - Across different types of observation spaces. For example, tabular (office grid world), RGB pixel observations (moving targets), and continuous vectors (safety gym).
> > >   - Across different types of action spaces: discrete actions (office grid world, moving targets) and continuous actions (safety gym)
> > > - Consider for example a robot arm manipulation scenario with 3 propositions (e.g each for putting an object in each of 3 bins), unknown continuous dynamics, vector observations, and vector actions. The safety gym tasks are representative of this scenario.
> > > - Finally, while we no longer have enough time left in the rebuttal period to include a manipulation task experiment, we hope this explanation is sufficient for the reviewer.
> > >
> > > **Generalisation**
> > > - We thought the reviewer's concern here was: because of the *applicability of LTL* concern above, there may be challenges in generalization across various scenarios (in the sense that our approach, which is focussed on temporal logic specified tasks, may not apply to various scenarios). Our citations were to address that concern. We apologise again if we misunderstood the reviewer's concern here.
> > > - Hence, can the reviewer please clarify what their *generalisation* concern is? We are dedicated to making our work as clear as possible.
> > >
> > > We thank the reviewer again for their engagement and for giving us the opportunity to elaborate more on our choice of tasks as we did take great care to use the most relevant ones for this paper. We look forward to the reviewer's thoughts on this point and the generalisation one.

---

> ### Comment · Reviewer_vdA3 · 2023-11-23
> **Thanks for the further explanation**
>
> Now I realize the applicability and generalizability of this framework. After considering the other reviewer's discussion, I believe this paper is valuable for the community. I hope to see more experimental results, such as manipulation task, in the camera-ready version if possible. I will raise my score to 6.

---

### Official Review · Reviewer_uyfa · 2023-11-02

**Soundness:** 3 good
**Presentation:** 3 good
**Contribution:** 3 good
**Rating:** 8
**Confidence:** 4

**Summary:**

This paper introduces the skill machines formalism. This is a variant of the reward machine formalisms where the agent learns Q-functions to satisfy individual task propositions, and then plans a temporal sequence of proposition states to be achieved through value iteration over the skill machine. Once the optimal path through the reward machine is found, the authors propose to use logical Q-function approximation from prior work to initialize the global semi-markov decision process Q-function. The initial policy can serve as a good zero-shot approximation to a satisficing policy. While the Q-function can also be optimized through exploration to generate a hierarchically optimal solution for the overall problem.

The key assumption in this problem is that all reward machine transitions are possible from all given reward machine states. Such as assumption is usually satsified when the domain involved proposition encoded as occupying certain regions of the state-space. It is definitely true if the propositions cover non-overlapping regions, but I am less certain if the assumption holds when there are overlapping regions as well. Nevertheless, the authors do clearly state the requisite assumption for the validity of their proposed approach.

**Strengths:**

**Originality, and significance**: The paper makes a significant contribution by providing an approach to compose skills both through proposition logic and temporal operators. This is quite a unique capability, and many works have proposed partial solutions for the same. The ideas, and evaluations presented in this work are quite compelling, and are theoretically sound. There are some claims that need to be examined in a greater detail as listed in the weakness and limitations sections, but nonetheless the paper is a significant advance.

**Evaluations**: The authors decision to push the complexity of the task specification must be commended. This evaluation explicitly tests for upward generalization capabilities. This can and should be strengthened by randomly sampling LTL goals from predefined distributions over specifications with a rejection sampling approach for unsatisfiable specifications, however, I am convinced regarding the validity of the approach from the theoretical arguments presented in the paper, and the scrupulous evaluations over varying complexity formulas.

**Comparison to prior LTL-based transfer approaches**: The comparison with prior approaches and the explanations of the key differences in capabilities is much appreciated.

**Weaknesses:**

**Environment validity:** The assumption of task satisfiability, and proposition transition reachability are quire strong. It is unclear what the test for whether an environment satisfies the assumptions required by the approach. The authors can ameliorate this by either providing domains where these assumptions always hold (environments where propositions encode visiting specific regions, and there being no overlap between propositions is one such environment), and they can also specifically search for a violating environment specification pair, and demonstrate what their approach outputs in that case. This information will be valuable to practitioners who would like to utilize skill machines.

**Relative expressivity of RMs vs LTL**: I believe that only a fragment of LTL can be translated into a reward machines, and there exist reward machines that cannot be expressed through an LTL formula. I would suggest the authors to appropriately restrict the LTL formula class they allow as input to the model to a fragment of LTL that is known to be expressible as a reward machine. (I believe the obligations segment)

**Questions:**

Please refer to the weakness section

---

> ### Author Response · Authors · 2023-11-20
>
> We really appreciate the reviewers' positive outlook on our paper, their time spent reviewing it, and providing useful feedback. We have updated the paper to reflect the reviewer’s suggestions regarding the weaknesses:
>
>
> **W1 [Environment validity]:**
> - We have added two experiments (Figure A5 and Figure A6 on page 18) where we evaluate the performance of our approach in different cases when the reachability assumption does not hold. Please see the highlighted text in Section A.4 for the discussion. In brief, we observe that the performance of the zero-shot agent may significantly decrease when the reachability assumption does not hold in some states (Figure A5), and may lead to complete failures in the worst-case scenario when the reachability assumption does not hold in all states (Figure A6). In both cases, we observe that the few-shot agent can still quickly improve on the zero-shot performance by learning the optimal policy.
> - It would be interesting to have an efficient test for whether the satisfiability and reachability assumptions hold for a given environment-specification pair. Unfortunately, this is non-trivial. Consider for example environments where propositions encode visiting specific regions and there is no overlap between propositions, as suggested by the reviewer. The Office GridWorld with a unique proposition for each decoration and coffee location is such an example. However, if the entrance to the office and mail rooms are blocked (say by a wall), then the office and mail propositions can never be satisfied when starting from positions outside of those rooms. That said, investigating efficient ways of checking those assumptions is an interesting avenue for future work.
>
>
> **W2 [Relative expressivity of RMs vs LTL]:**
> - Indeed, only some fragments of LTL can be translated into reward machines. We restrict ourselves to such fragments, such as co-safe LTL as shown in Camacho et al. (2019). We have stated this more explicitly in Section 2.1 page 3. We thank the reviewer for this suggestion.

---

> > ### Comment · Reviewer_uyfa · 2023-11-22
> > **Happy with modifications**
> >
> > I retain my strong positive evaluation of this paper. Having read the other reviewer's comments, I think the concerns are as follows:
> > 1) Comparisons with prior works in transfer of LTL formulas to novel tasks: I largely agree with the author's assessments that HRM and CRM are conceptual standins for prior approaches as mapped in their rebuttal, and while there might be implementation details, requiring the authors to run them would be running baselines for the sake of running baselines. However it is important that the authors clarify the prior approaches and their mappings to the baselines in the sections where they define these baselines. Running these baselines explicitly also helps to provide direct empirical evidence that they are conceptually similar to the selected baselines, but again I do not feel that this should be a disqualifying criterion.
> > 2) Choice of LTL as a specification language: I believe that utility of LTL as a specification tool is an open question with no settled answers. Whether scalar rewards or STL are more beneficial as specification frameworks for RL is outside the scope of the discussion of this paper. As a reviewer my belief about LTL or Reward machines not being an appropriate tasks specification should not be the basis of assessing the soundness of the paper.
> >
> > The authors have also tried to respond to the soundness and clarity concerns throughout their responses, and I believe that there are no outstanding correctness concerns. The remaining clarity concerns can be addressed without substantially changing the rhetorical claims of the paper.
> >
> > In view of this I retain my positive assessment, and invite the other reviewers to revisit their assessments.

---

> > > ### Author Response · Authors · 2023-11-22
> > > **Response to Reviewer uyfa**
> > >
> > > We thank the reviewer once again for their time, consideration and overall support of our work.
> > >
> > > We will certainly ensure that the prior approaches raised during this discussion period are discussed in Section 4 where we describe our experimental design and methodology. This will include the point of how these similar baselines relate to the benchmarks which we do use in our experiments. Finally, we are glad to see that we have addressed any soundness concerns and that there are no outstanding correctness concerns. We will certainly use the reviewer’s cogent review and suggestions to further improve the clarity of our paper for the subsequent revision.

---

### Author Response · Authors · 2023-11-20
**General Comment**

We would like to once again thank all of the reviewers for their time and constructive feedback on our paper.


We note that one of the primary weaknesses raised by Reviewer 38Qb and Reviewer pDWg was our selection of benchmark algorithms. This is an extremely important point and one we take very seriously. We thank these reviewers for raising the topic and thought we would use this as an opportunity to elaborate more for all reviewers as we did take great care in our choice of benchmarks. We hope this would further contextualize our work in the broader literature and justify our experimental design. **To be specific, Reviewer 38Qb recommended comparing with the Logical Options Framework (LOF) (Araki et al., 2021) and LTL2Action (Vaezipoor et al., 2021). Reviewer pDWg instead recommends that we compare against Luo et al. (2019) and León et al. (2019)**.


To summarize our position, **we believe that HRM (Icarte et al., 2022) is representative of LOF (Araki et al., 2021) and that CRM (Icarte et al., 2022) is representative of LTL2Action (Vaezipoor et al., 2021) and León et al. (2019).**


 - While approaches that only leverage temporal skill composition like HRMs and LOF are able to learn tasks fast, they all still require learning the sub-task options for most new tasks. In LOF, options are learned for each subgoal proposition of the reward machine, and then value iteration or Q-learning is used to learn a meta-policy over options to solve the **current** task. These options can then be reused in new tasks (i.e new reward machines) where the same sub-goal propositions appear. With both HRMs and LOF the options learned from a previous task cannot be transferred satisfactorily to some new tasks (Liu et al. 2022). For example in the office world, if in a previous task the agent learns an option for “getting coffee”, it can reuse it to “get coffee then deliver it to the office”, but it cannot reuse it in a new task where it needs to “get coffee without breaking decorations”. In contrast, Skill Machines (our work) are able to bypass this issue by learning skill primitives that can then be composed zero-shot to get coffee with or without constraints. Thus, HRMs and LOF omit spatial compositionality and **HRMs provide an appropriate baseline which is representative of the limitations of all such algorithms without spatial compositionally that instead leverages temporal compositionally to learn hierarchically-optimal policies**.

  -  Similarly, LTL2Action (Vaezipoor et al., 2021) and León et al. (2019), assume that the tasks are specified using LTL and TTL respectively. LTL2Action propose using a separate LTL module (a graph neural network) to end-to-end learn an embedding of “LTL progressions”. They then demonstrate empirically that their approach converges to more optimal policies than a myopic baseline and also achieves comparable or better generalisation. León et al. (2019) propose instead to use a combined or separate task and observation module (LSTMs and CNNs) to end-to-end learn an embedding of “TTL progressions”. They then demonstrate empirically that their approach can learn given tasks and also achieve some generalisation to new ones. However, LTL2Action and León et al. (2019) do not leverage spatial or temporal skill composition. This is also true of CRMs (Icarte et al., 2022). Hence, all three algorithms rely on function approximation (e.g neural networks) to generalise both spatially and temporally. Considering the spatial and temporal curses of dimensionality described in our paper (and the arbitrarily long temporal sequence of sub-tasks present in each task), they have to learn every new task with high probability. Consequently, **CRMs provide a baseline representative of the class of algorithms without spatial or temporal composition that instead leverages the compositional structure of LTL or RMs to learn optimal policies efficiently**.

 - Thus, while we acknowledge the important contribution of all such prior works, we have taken great care in choosing a reflective set of baselines which will be of broadest interest possible. Thus, we are uncertain what benefit a comparison with LOF, LTL2Action or León et al. (2019) would provide which is not already achieved by comparing with HRMs and CRMs. We would appreciate all reviewers’ thoughts on this point.

Finally, we note that Luo et al. (2019) assumes the usual planning/optimal-control setting, which is a very different setting to ours and the prior works cited in this paper. Mainly, they assume that the environment dynamics are known and amenable to planning. Note that in our work and the relevant prior works, the environment dynamics are not known (hence why RL is used) and in our case planning only happens at the level of the reward machines.

---

> ### Author Response · Authors · 2023-11-20
> **General Comment (cont.)**
>
> We once again thank Reviewer 38Qb and Reviewer pDWg for raising this important discussion which we have not yet had the opportunity to address. We hope by raising this as a general comment we better contextualize our work in general and maintain the transparency of our experimental design and reasoning.
>
>
> Icarte et al. "Reward machines: Exploiting reward function structure in reinforcement learning." JAIR 2022.
>
> Vaezipoor et al. "Ltl2action: Generalizing ltl instructions for multi-task rl." ICML 2021.
>
> Araki et al. "The logical options framework." CoRR 2021
>
> León et al. "Systematic generalisation through task temporal logic and deep reinforcement learning." arXiv preprint  2020.
>
> Luo et al. "Transfer Planning for Temporal Logic Tasks." CDC 2019.

---

### Meta-Review · Area_Chair_iG4N · 2023-12-06

**Metareview:**

The paper introduces a finite state machine, called skill machine, which enable agents to tackle complex tasks involving temporal and spatial composition. The method guarantees that the learnt policy aligns with the logical task specification, provided in a LTL fragment. The authors have incorporated various suggestions from the reviewers that has helped provide clarity about paper's central contribution, theoretical guarantees, and its relationship to previous work.

**Justification For Why Not Higher Score:**

The concept of skill machines seems like a reasonable extension of reward machines and not a ground breaking technique in itself. There are also some assumptions about the underlying transitions of the MDP that may restrict applicability. A theoretical bound on the sub-optimality of learned policies would have provided more confidence in the proposed method.

**Justification For Why Not Lower Score:**

The proposed technique provides a novel mechanism to leverage compositionality for complex tasks and extends the applicability of such techniques to more complex environments and tasks.

---

### Decision · Program_Chairs · 2024-01-16

Accept (poster)